



# Quantifying permafrost ground ice contents in the Tien Shan and Pamir (Central Asia): A Petrophysical Joint Inversion approach using the Geometric Mean model

Tamara Mathys[1], Muslim Azimshoev[2], Zhoodarbeshim Bektursunov[4], Christian Hauck[1], Christin Hilbich[1], Murataly Duishonakunov[7], Abdulhamid Kayumov[3], Nikolay Kassatkin[6], Vassily Kapitsa[6], Leo C.P. Martin[5], Coline Mollaret[1], Hofiz Navruzshoev[3,8], Eric Pohl[1], Tomas Saks[1], Intizor Silmonov[2], Timur Musaev[4], Ryskul Usubaliev[4], and Martin Hoelzle[1]

[1]Department of Geosciences, University of Fribourg, Fribourg, Switzerland
[2]Aga Khan Agency for Habitat (AKAH), Tajikistan
[3]Center for Research of Glaciers of the Academy of Sciences of Tajikistan, Tajikistan
[4]Central Asian Institute for Applied Geosciences (CAIAG), Kyrgyzstan
[5]Aix Marseille Univ, CNRS, IRD, INRAE, CEREGE, Aix-en-Provence, France
[6]Central Asian Regional Glaciological Centre Under the Auspices of UNESCO, Almaty, Kazakhstan
[7]Kyrgyz National University, Kyrgyzstan
[8]Mountain Societies Research Institute of the University of Central Asia, Tajikistan

**Correspondence:** Tamara Mathys (tamara.mathys@unifr.ch)

**Abstract.** In the Central Asian Tien Shan and Pamir mountain ranges, permafrost is extensive, but in-situ data on permafrost remains scarce. Quantitative analysis of permafrost's subsurface components—ice, water, air, and rock—is vital for not only discerning the impact of climate change on increased slope instability due to permafrost degradation, but also for understanding its role as a potential water resource in high-altitude environments. Recent studies have employed a Petrophysical Joint

Inversion (PJI) approach combining geoelectrical and seismic refraction data to model the subsurface's four phases (fractions of air, water, ice, and rock). However, most of these studies primarily rely on Archie's law, which has limitations in coarse blocky substrates typical of mountainous terrains. Recognizing this limitation, the electrical Geometric Mean (PJI-GM) model may be used as an alternative implementation within the PJI. In this study, we assess the suitability of using the PJI-GM model across an extensive geophysical dataset comprising 22 profiles in Central Asia (Kyrgyzstan and Tajikistan). Our goals are to (i)

address the existing data gap concerning mountain permafrost and ground ice contents in the Tien Shan and Pamir of Central Asia and (ii) evaluate the performance of the PJI-GM model in comparison to Archie's law within the PJI framework across the different landforms at remote sites. The findings reveal that the ground ice content is more specific to landform types than to the different geographic regions surveyed, with rock glaciers exhibiting the highest mean ice contents (38-60 %), followed by moraines (18-40 %), talus slopes (20-40 %), and fine-grained sediments (0-20 %). The PJI-GM model performed especially

well for ice-rich landforms such as rock glaciers, accurately reflecting high ice contents with minimal variability between different model runs. The quality of a model result was hereby assessed by comparing a multitude of different model runs with different sets of inversion parameters and petrophysical variables using a clustering approach. This research provides one of the first comprehensive (geophysical) in-situ datasets on permafrost on various landforms and sites in Central Asia, highlighting





the potential of the PJI-GM model as a more suitable alternative to Archie's law, particularly for rock glaciers and other ice-rich landforms. These findings significantly advance our understanding of permafrost in the Tien Shan and Pamir and serve as a baseline dataset for future modeling studies.

## 1 Introduction

With ongoing climate change, permafrost, which is defined as subsurface material with temperatures at or below 0 °C for at least two consecutive years, is experiencing increased regional and global warming and degradation (Biskaborn et al., 2019; Etzelmüller et al., 2020). For mountain permafrost, the importance of this degradation lies two-fold. First, permafrost degradation affects slope stability, thus compromising the structural integrity of mountain slopes, leading to increased landslides, debris flows, and altered sediment transport, ultimately escalating geohazard risks (Daanen et al., 2011; Haeberli et al., 2017; Ravanel et al., 2017). Furthermore, the degradation of mountain permafrost raises critical questions about its role in the hydrological cycle. Firstly, there's the question of whether thawing permafrost acts as a significant water resource, and how this contribution might evolve over time (e.g. Jones et al., 2019; Luo et al., 2020). Secondly, permafrost thaw can also influence a catchment's evaporation and runoff patterns, adding complexity to predicting future water resources (Martin et al., 2023). While its potential as an alternative water resource, particularly in dry regions, has been noted, its role in seasonal water supply and long-term river system sustainability remains unclear (Arenson and Jakob, 2010; Arenson et al., 2022; Amschwand et al., 2024). Comprehensive baseline data on the distribution, thermal state, and ground ice contents of permafrost is essential for assessing potential climate change impacts on slope stability and hydrology, and mitigating risks associated with geohazards. However, despite the expected widespread occurrence of permafrost across Central Asia (Marchenko et al., 2007), a considerable data gap exists for the Tien Shan and Pamir mountain ranges. Here, permafrost observations are limited and localized in the Northern and Central Tien Shan, while in the Pamir and Pamir Alay only sporadic observations are available (Barandun et al., 2020; Hoelzle et al., 2019). This scarcity of data underscores the urgent need for comprehensive in-situ investigations to characterize permafrost distribution, thermal state, and ground ice content in the Tien Shan and Pamir mountain ranges.

In-situ methods are crucial for effectively detecting and monitoring changes in the thermal state and ground ice contents of mountain permafrost, as most permafrost features are not directly detectable with remote sensing techniques. Permafrost is typically monitored using a combination of in-situ methods such as boreholes, geophysical measurements and ground surface temperature (GST) loggers (e.g. PERMOS, 2023). Boreholes enable direct measurements of ground temperatures for thermal analysis, and drill cores can provide information about ground ice contents, thereby yielding accurate data on the subsurface characteristics of permafrost (Noetzli et al., 2021). In Central Asia, despite the current lack of data, extensive research on permafrost, focusing on borehole drillings, began early. The systematic study of permafrost, in particular in the Northern and Central Tien Shan of Kyrgyzstan and Kazakhstan, was initiated in the 1950s (e.g. Ermolin et al., 1989; Gorbunov, 1967, 1970), but Duishonakunov (2014) noted that there have been permafrost observations in this region even earlier. Until the 1990s, numerous boreholes were drilled in this region (Marchenko et al., 2007). A lower boundary for sporadic permafrost in the Tien Shan was estimated to be at an altitude of 2800 – 3000 m a.s.l (Gorbunov et al., 1996). Marchenko et al. (2007) found an



increase in permafrost temperatures in the Tien Shan within the range of 0.3 °C to 0.6 °C using observations with an average increase of the active layer by about 23 % from the 1970s to 2004. Similarly, Seversky (2017) detected a slight increasing trend in ground temperatures between 1995 and 2016 (0.01 °C per year from 1974 to 2016) in boreholes in the Northern Tien Shan. While this historical basis provides valuable context, accessing the datasets can be challenging and only one borehole in the Northern Tien Shan is still actively observed today within the GTN-P network (Seversky, 2017). Furthermore, borehole drilling is expensive, logistically challenging, and can only provide point information.

Geophysical techniques such as Electrical Resistivity Tomography (ERT), Refraction Seismic Tomography (RST), or Ground Penetrating Radar (GPR) are invaluable for expanding our understanding of permafrost over larger areas. They can, for example, be used to delineate the active layer and taliks, as well as assess changes in subsurface properties if measurements are repeated regularly (e.g. Hauck and Kneisel, 2008; Hilbich et al., 2009; Monnier and Kinnard, 2013; Mollaret et al., 2019; Halla et al., 2021; Mollaret et al., 2020; Kneisel et al., 2008; Vonder Mühll et al., 2001; Boaga et al., 2020; Herring et al., 2023). In Central Asia, geophysical investigations are limited to the Central and Northern Tien Shan (Seversky, 2017). However, spatial and temporal coverage is relatively sparse. ERT surveys were conducted in the Northern Tien Shan in 2013 and in 2017 (Galanin et al., 2017), where they found high resistivities typical for (ice-rich) permafrost. Bolch et al. (2019) used GPR measurements to estimate ground ice contents of ice-debris complexes in the Central Tien Shan and mapped a total of 74 rock glaciers and ice-debris complexes using remote sensing. Boreholes have revealed the presence of ground ice in the form of ice lenses in moraines (Marchenko et al., 2007). Bolch and Marchenko (2009) estimated rock glacier ground ice contents in the Tien Shan based on an empirical relationship proposed by Brenning (2005). However, quantitative estimates of ground ice contents based on field measurements are, to our knowledge, not yet available for the Tien Shan. Furthermore, most remote sensing research on permafrost in the area is focused on rock glaciers and is often lacking in-situ data for validation (Blöthe et al., 2019; Sorg et al., 2015; Kääb et al., 2021; Bertone and Barboux, 2020).

Data and research on permafrost in general are much more scarce in the Pamir and the Pamir Alay (Barandun et al., 2020). Here, permafrost occurrence has been described down to an altitude of 3800 m a.s.l. Climate model projections for the extended Tibetan Plateau, which in most studies includes parts of the Pamir, suggest a reduction of near-surface permafrost of 39% by 2050 and up to 81% by 2100 (Bolch et al., 2019). Most current studies that focus on permafrost in the Pamir rely on remote sensing and are not focusing on ground ice contents but rather concentrate on hazards associated with potentially increasing slope instability as permafrost degrades (Jones et al., 2021b; Mergili et al., 2013; Mergili and Schneider, 2011). To our best knowledge, no prior geophysical investigations on permafrost have been conducted in the Pamir region to this date.

Yet, understanding permafrost subsurface conditions, such as volumetric contents of ice, water, and rock is crucial for developing a comprehensive understanding of permafrost processes and for evaluating associated degradation risks. Permafrost genesis in mountainous regions is influenced by a combination of climatic, geological, and environmental factors (Gilbert et al., 2016). The formation of ground ice within permafrost can occur through processes such as the freezing of infiltrating precipitation, the migration of water to freezing fronts, the burial of glacier ice, and the burial of snow (e.g. Pollard, 1990; Bockheim and Tarnocai, 1998; Monnier et al., 2011; Kenner et al., 2017; Gilbert et al., 2016). These processes vary across different landforms, leading to diverse permafrost characteristics and ground ice contents. However, quantifying ground ice





content within different landforms remains a significant challenge. For example, rock glaciers have been shown in numerous studies to contain significant amounts of ground ice (e.g. Vonder Mühll and Holub, 1992; Vonder Muhll and Haeberli, 1990; Krainer et al., 2015; Monnier et al., 2011). However, even for relatively well-studied landforms like rock glaciers, most esti-

mates rely on empirical relationships to estimate ground ice contents (Brenning, 2005), with only a limited number of studies providing quantitative measurements (e.g. Halla et al., 2021; Pavoni et al., 2023; Hilbich et al., 2022; Mollaret et al., 2020). This lack of direct measurement data and the complexity of permafrost subsurface conditions make it difficult to establish a consensus on typical ice content ranges, particularly for landforms beyond rock glaciers such as talus slopes or moraines. To our knowledge, no single reference paper comprehensively summarizes expected ground ice contents across various mountain

permafrost landforms.

However, in recent years, various (geophysical) approaches have been employed to quantify these subsurface constituents, in particular the ground ice contents. Hauck et al. (2011) introduced the Four Phase Model (4PM), which integrates electrical resistivity and P-wave velocity measurements to characterize water, ice, air, and rock contents in the subsurface. This model has been used in various studies, and results have been shown to fit well with ground truth data such as borehole data or field

observations (e.g. Halla et al., 2021; Hilbich et al., 2022; Kunz et al., 2022). The 4PM uses Archie's Law (Archie, 1942) to relate the bulk resistivity of a material to its porosity and the resistivity of the pore water, which works well in environments where electrolytic conduction dominates. However, Archie's Law does not directly account for the fractions of air and ice. In the 4PM, these fractions are indirectly constrained by integrating ERT data with RST data, which provides additional information on the seismic properties of the subsurface materials. Wagner et al. (2019) further developed a Petrophysical Joint Inversion (PJI)

scheme, that builds on the principles of the 4PM but jointly inverts the electrical and seismic data. The PJI has been applied in multiple studies (e.g. Mollaret et al., 2020; de Pasquale et al., 2020; Pavoni et al., 2023; Steiner et al., 2021b; Klahold et al., 2021). Mollaret et al. (2020) tested different petrophysical models within the PJI, and suggested that the so-called electric Geometric Mean Model (hereafter referred to as PJI-GM) could offer more realistic results compared to the commonly used Archie's law, which is currently the main resistivity equation implemented in the PJI. This stems from the recognition that

Archie's law (hereafter referred to as PJI-AR) is generally considered valid when electrical conduction through fluids within the pore space dominates over conduction through the solid matrix itself, a condition that is not universally justified. The Geometric Mean Model assumes that the modeled space is composed of a mixture of the four phases (rock, ice, water, and air), with each phase being randomly distributed within the subsurface. This approach allows for the inclusion of the fractions of all four phases, as opposed to Archie's law. However, the PJI-GM model presents challenges as additional unknowns (the

resistivities of ice, air and rock in addition to the pore water resistivity that is needed in Archie's law) are added to the system of equations. Furthermore, securing model convergence within the PJI-GM can be challenging, as indicated by substantial errors in numerous model outputs (Mollaret et al., 2020). Finally, the PJI-GM has not yet been tested extensively for a large dataset comprising multiple examples of different landforms.

Since 2021, we have been addressing the in-situ data gap on permafrost in Central Asia by conducting extensive geophysical

surveys. These surveys involved ERT and RST measurements at various study sites in the Tien Shan and Pamir mountain ranges. In this study, we present these data and assess the suitability of using the PJI-GM model within the PJI framework





to estimate ground ice content distribution across an extensive geophysical dataset comprising 22 profiles in Central Asia (Kyrgyzstan and Tajikistan). Our research encompasses diverse landforms, including moraines, rock glaciers, talus slopes and fine-grained sediments. Our goals are to (i) address the existing data gap concerning mountain permafrost and ground ice

contents in the Central Asian region and (ii) evaluate the performance of the Geometric Mean Model in comparison to Archie's law across different landforms. The baseline dataset established within this study is essential for developing accurate models and predictions of future permafrost dynamics and their associated impacts on hydrology and geohazards in the face of climate change and may assist local populations in adapting to forthcoming changes by informing water resource management and disaster risk reduction policies.

## 130 2 Study sites

Geophysical measurements were carried out at 10 different sites distributed across the Pamir and Tien Shan mountain ranges and ranging from 3100 to 4580 m a.s.l. A total of 38 ERT profiles and 22 RST profiles were measured on different landforms which were categorized into rock glaciers (RG), talus slopes (TS), fine-grained sediments (SED), and moraines (MO). Figure 1 shows the location of each study site with pictures of the different landforms investigated. The climate in the Central Asian

region is mostly semi-arid, with high seasonal variability due to its continental location, and considerable regional variations (e.g. Aizen et al., 2009; Haag et al., 2019; Barandun and Pohl, 2023). The sites chosen for the permafrost analyses reflect different climatic and geomorphological settings. They are part of a comprehensive cryospheric monitoring network being established in the region, covering all cryospheric variables (snow, glaciers, permafrost). Meteorological stations were installed near most of the study sites as part of projects to re-establish the glacier monitoring network from past and ongoing projects

efforts (Hoelzle et al., 2017; Schöne et al., 2013; Zech et al., 2021). Mean annual air temperatures (MAAT) and mean annual precipitation were calculated and their inter quartile ranges (IQR) are provided in Tab. 1. Table 1 provides a summary of the study sites, highlighting variations in climate and other relevant site information. A short introduction of each sub-region (after Barandun and Pohl (2023) and Zandler et al. (2016)) and corresponding study sites is given in the following sections. Figure 2 shows a zoom into each study site, showing the location of all geophysical profiles (ERT and RST). Data acquisition details

for each geophysical profile are given in Table 2.

### 2.1 Northern Tien Shan

The Northern and Northwestern Tien Shan sub-region is characterized by a comparatively moist climate with precipitation rates of about 700 mm $a-1$ (e.g. Aizen et al., 2006; Barandun et al., 2018; Guan et al., 2022). Earlier studies have identified a significant number of large rock glaciers in the region (Blöthe et al., 2019; Bertone et al., 2019). The study sites Golubin and

#599 are located within this sub-region. MAAT at the Golubin site (2014 - 2021, at an altitude of 3305 m a.s.l.) is approximately -1.47 °C (see Fig. 1). The profiles located at the Golubin study site in the Ala Archa catchment (about 30 km from the capital of Kyrgyzstan, Bishkek) range from 3050 m a.s.l to 3410 m a.s.l and include all four landform categories. Study site #599 is



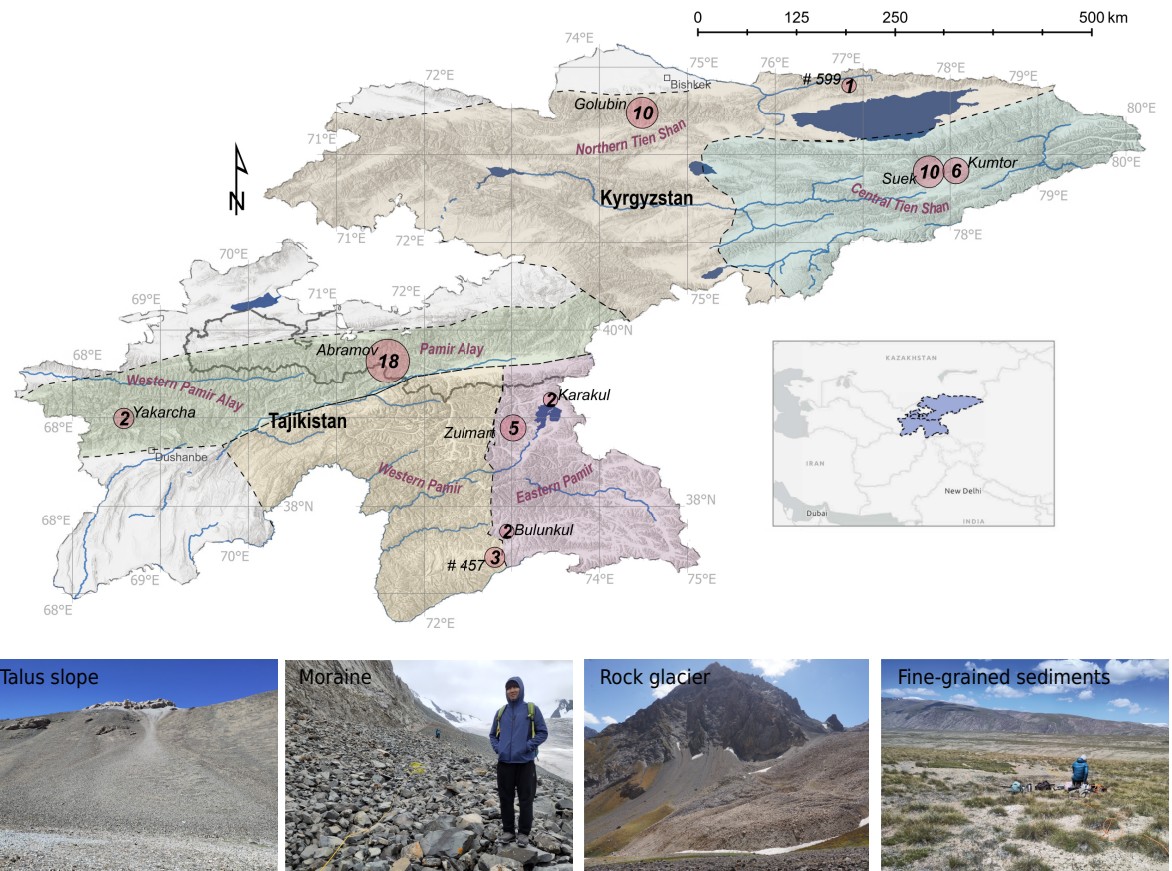

**Figure 1.** Study sites are marked with pink circles, with diameters proportional to the number of geophysical profiles. Each circle is labeled with its study site name and includes the number of conducted ERT and RST profiles. Different sub-regions are distinguished by different colors. The pictures display sampled landforms such as talus slopes, moraines, rock glaciers, and fine-grained sediments. Sources of the background maps: Esri, DigitalGlobe, GeoEye, i-cubed, USDA FSA, USGS, AEX, Getmapping, Aerogrid, IGN, IGP, swisstopo, and the GIS User Community

located to the North of lake Issykul. The ERT profile measured at this site is located on a moraine at an altitude of 3780 m a.s.l. Unfortunately, no meteorological station is available in proximity to this site.

## 2.2 Central Tien Shan

Compared to the Northern Tien Shan, the climate in the Central Tien Shan is drier with annual precipitation amounts of about 350 mm $a^{-1}$ (Aizen et al., 2006). The study sites Kumtor and Suek are located within this sub-region south of Lake Issykul. Both study sites are located on a high-mountain plateau (mean of 3600 m a.s.l.) in the Upper Naryn catchment. The profiles measured at the Kumtor site are all located on fine-grained, vegetated sediments, about 5 km from the Kumtor gold



**Table 1.** Study site overview: MAAT and mean annual precipitation were calculated from available meteorological stations (MS) close to the geophysical measurement locations, where available and as described in (Hoelzle et al., 2017) and (Schöne et al., 2013). The time span for the calculation is indicated in parentheses next to the MS name. Wherever this was not possible, the information was taken from other studies and are indicated in the references column.

| Study site | Region | MAAT (° C) | Precipitation (mm a$^{-1}$) | Mean altitude of profiles (m a.s.l) | ERT profiles | RST profiles | References |
|---|---|---|---|---|---|---|---|
| Abramov | Pamir Alay | -4.6 | 750 | 3850 | 10 | 8 | Abramov MS (2011-2021) Hoelzle et al. (2017); Kronenberg et al. (2020) |
| Golubin | Northern Tien Shan | -1.47 | 700 | 3185 | 7 | 3 | Golubin MS (2014-2021); Schöne et al. (2013); Aizen et al. (2006) |
| Suek | Central Tien Shan | -6.84 | 290 | 3722 | 7 | 3 | Tien Shan MS (1963 - 2010); Machguth et al. (2023) |
| Kumtor | Central Tien Shan | -6.84 | 290 | 3540 | 3 | 3 | Tien Shan MS (1963 - 2010); Machguth et al. (2023) |
| #599 | Northern Tien Shan | - | - | 3780 | 1 | - | No meteorological data available |
| Yakarcha | Western Pamir Alay | -2.14 | 350 | 3300 | 2 | - | Yakarcha MS (2019 - 2023); Hoelzle et al. (2017) |
| #457 | Western Pamir | -1.11 | - | 4580 | 2 | 2 | Jelondy MS (2020 - 2022) |
| Zulmart | Eastern Pamir | -4.1 | - | 4560 | 3 | 2 | Zulmart MS (2021- 2023); Hoelzle et al. (2017) |
| Bulunkul | Eastern Pamir | -5.38 | 110 | 3720 | 1 | 1 | Bulunkul MS (1960-2017), data provided by local partners |
| Karakul | Eastern Pamir | -3.7 | 80 | 4210 | 1 | 1 | Karakul MS (1934-2017), data provided by local partners |

mine. Several boreholes were drilled at this location in the 1980s (Marchenko et al., 2007) but are mostly inactive today. A new borehole was drilled in 2022 at the location of profile KUM04, revealing frozen conditions, a shallow active layer of approximately 1 - 1.5 m, and saturated ground ice conditions in the upper part of the drill core. The profiles measured at the Suek study site cover all four landform types and are distributed along the Suek pass with altitudes ranging from 3400 - 3880 m a.s.l.

**2.3  Pamir Alay**

The Pamir Alay meridionally separates the the Pamir and Tien Shan mountains. It is located in the northwest of the Pamir mountain range and encompasses different mountain ranges spanning from southern Kyrgyzstan to the northwest of Tajikistan. The study sites Abramov (KG) and Yakarcha (TJ) are located in this sub-region. At the Abramov study site, located in Vakhsh catchment, MAAT at an altitude of 4100 m a.s.l is -4.6 °C (measurements from 2010 - 2021). Typical precipitation rates in

this part of the Pamir Alay are around 750 mm $a^{-1}$ (Barandun et al., 2015; Kronenberg et al., 2020). The geophysical surveys include various rock glaciers, talus slopes, fine-grained sediments, as well as multiple measurements on the Abramov glacier



**Table 2.** Data acquisition parameter and onsite permafrost observations. The profiles which have both ERT and RST data are marked in bold. RST profiles are usually shorter than the ERT profiles, but we used the same spacing for both methods. The lines are shown in Fig. 2. ERT configurations: W = Wenner, DD = Dipole-Dipole. Landform classes: RG = Rock glacier; MO = moraine; TS = talus slope; SED = fine-grained sediments.

| Profile | Acquisition date | Site | length (m) | ERT array | spacing (m) | landform | mean altitude (a.s.l) | Geology | PF observations |
|---|---|---|---|---|---|---|---|---|---|
| abra01 | 2021-08-20 | Abramov | 235 | W | 5 | MO | 3764 | - | - |
| **abra02** | **2021-08-21** | **Abramov** | **595** | **W** | **5** | **RG** | **3867** | **Limestone blocks** | **furrows and ridges** |
| **abra03** | **2021-08-22** | **Abramov** | **475** | **W** | **5** | **RG** | **3890** | **Diamict** | **furrows and ridges** |
| **abra04** | **2021-08-23** | **Abramov** | **355** | **W** | **5** | **RG** | **3909** | **Diamict** | **furrows and ridges, (massive) ice outcrops visible** |
| **abra05** | **2021-08-23** | **Abramov** | **235** | **W** | **5** | **TS** | **3874** | **Diamict** | **-** |
| abra06 | 2021-08-24 | Abramov | 235 | W | 5 | TS | 3890 | - | - |
| **abra07** | **2021-08-24** | **Abramov** | **235** | **W** | **5** | **SED** | **3851** | **Diamict** | **-** |
| abra08 | 2021-08-24 | Abramov | 235 | W | 5 | MO | 3774 | - | ice outcrops in moraine |
| **abra09** | **2022-07-26** | **Abramov** | **355** | **W** | **5** | **RG** | **3873** | **Limestone diamict** | **furrows and ridges** |
| **abra10** | **2022-07-24** | **Abramov** | **235** | **W** | **5** | **MO** | **3817** | **Coarse diamict** | **ice outcrops in moraine** |
| **GOL01** | **2021-07-18** | **Golubin** | **235** | **W / DD** | **5** | **TS** | **3230** | **Granitic blocks** | **-** |
| GOL02 | 2021-07-19 | Golubin | 235 | W | 5 | RG | 3130 | - | furrows and ridges |
| GOL03 | 2021-07-19 | Golubin | 235 | W | 5 | RG | 3122 | - | furrows and ridges |
| GOL05 | 2022-08-08 | Golubin | 235 | W / DD | 5 | MO | 3410 | - | - |
| GOL06 | 2022-08-08 | Golubin | 235 | W | 5 | RG | 3192 | - | furrows and ridges |
| **GOL07a** | **2022-08-09** | **Golubin** | **235** | **W** | **5** | **SED** | **3200** | **Granitoids** | **-** |
| **GOL07b** | **2022-08-10** | **Golubin** | **235** | **W** | **5** | **MO** | **3200** | **Diamict** | **-** |
| **GOL07c** | **2022-08-10** | **Golubin** | **595** | **W / DD** | **5** | **RG** | **3155** | **Diamict** | **furrows and ridges** |
| **SUE01** | **2021-08-02** | **Suek** | **213** | **W / DD** | **3** | **TS** | **3835** | **Gabbro association** | **Gelifluction patterns** |
| **SUE02** | **2021-08-02** | **Suek** | **235** | **W / DD** | **5** | **TS** | **3854** | **Gabbro association** | **Gelifluction patterns** |
| SUE03 | 2021-08-03 | Suek | 235 | W | 5 | RG | 3524 | - | furrows and ridges |
| SUE03_V | 2021-08-03 | Suek | 235 | W | 5 | TS | 3501 | | in lower part of profile |
| SUE04 | 2021-08-03 | Suek | 235 | W | 5 | SED | 3420 | - | - |
| **SUE05** | **2023-07-12** | **Suek** | **235** | **W / DD** | **5** | **MO** | **3962** | **Gabbro association** | **ice found when digging** |
| SUE06 | 2023-07-12 | Suek | 235 | W / DD | 5 | MO | 3960 | - | ice found when digging |
| yak01 | 2022-08-28 | Yakarcha | 235 | W | 5 | RG | 3381 | - | furrows and ridges |
| yak02 | 2022-08-29 | Yakarcha | 235 | W | 5 | RG | 3388 | - | furrows and ridges |
| **ZUL01** | **2023-08-09** | **Zulmart** | **355** | **W / DD** | **5** | **RG** | **4575** | **Limestone-schist diamict** | **furrows and ridges** |
| ZUL02 | 2023-08-10 | Zulmart | 235 | W / DD | 5 | MO | 4584 | - | - |
| ZUL03 | 2023-08-11 | Zulmart | 235 | W | 5 | SED | 4537 | - | - |
| **KAR01** | **2023-08-07** | **Karakul** | **235** | **W / DD** | **5** | **SED** | **4231** | **Diamict** | **-** |
| **no457_01** | **2023-08-16** | **no457** | **235** | **W / DD** | **5** | **SED** | **4526** | **Diamict/Fluvial sediments** | **-** |
| no457_02 | 2023-08-17 | no457 | 235 | W | 5 | RG | 4641 | - | furrows and ridges, ice visible between blocks |
| **BUL01** | **2023-08-05** | **Bulunkul** | **141** | **W / DD** | **3** | **SED** | **3720** | **Lake sediments** | **-** |
| **KUM01** | **2022-08-17** | **Kumtor** | **235** | **W / DD** | **5** | **SED** | **3537** | **Diamict** | **-** |
| **KUM02** | **2022-08-17** | **Kumtor** | **835** | **W / DD** | **5** | **SED** | **3525** | **Diamict** | **-** |
| **KUM04** | **2022-08-19** | **Kumtor** | **235** | **W / DD** | **5** | **SED** | **3552** | **Diamict** | **Borehole confirms permafrost and saturated ice conditions in uppermost layers** |
| no599_01 | 2021-07-28 | no599 | 235 | W | 3 | MO | 3756 | - | - |





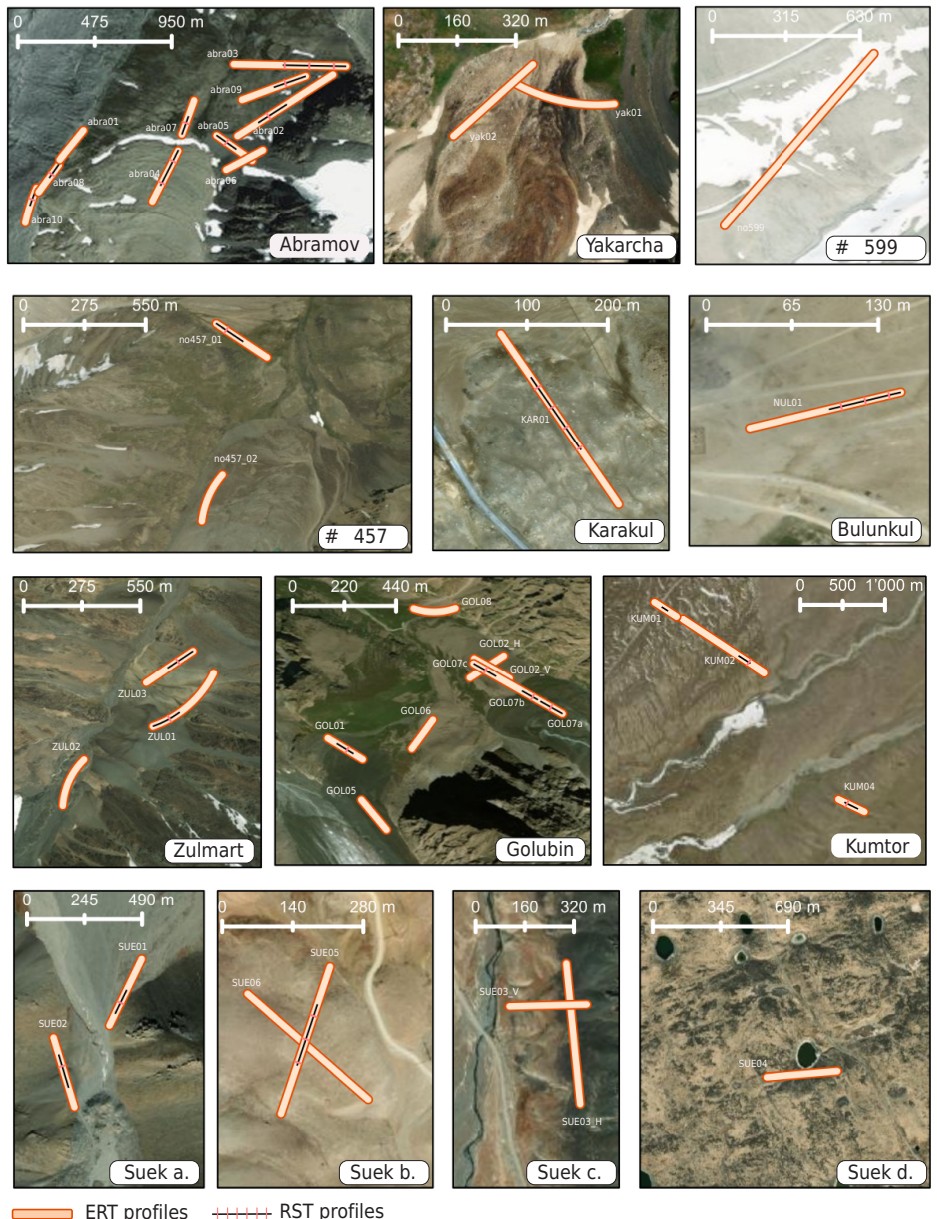

**Figure 2.** Distribution of geophysical profile lines across each study site. Each line is labeled with the corresponding profile name, which can be cross-referenced with detailed information provided in Table 2. The profile lines are depicted to illustrate the spatial extent and orientation of the geophysical surveys conducted. Sources of the background maps: Esri, DigitalGlobe, GeoEye, i-cubed, USDA FSA, USGS, AEX, Getmapping, Aerogrid, IGN, IGP, swisstopo, and the GIS User Community.

lateral moraine. At the Yakarcha study site, located to the north of Dushanbe in the Western part of the Pamir Alay in the Varzob





catchment, MAAT is around -2.14 °C (measurements from 2019 - 2023) at an altitude of around 3500 m a.s.l. Precipitation in the region (measured at nearby Ansob pass) is around 400 mm a$^{-1}$ (Rahmonov et al., 2017). Here, ERT profiles were measured on a large rock glacier.

## 2.4 Western Pamir

The Western Pamir is characterized by deeply incised valleys and high mountain ranges (5000 - 6000 m a.s.l.) (Breu et al., 2003). The climate is mainly influenced by the Westerlies with minimal rainfall in the summer months (Aizen et al., 2009; Barandun and Pohl, 2023). The Western Pamir shows extreme precipitation differences at regional to local scales, e.g. with the highest monitored long-term mean of 2234 mm $a^{-1}$ at Fedchenko Glacier (4300 m a.s.l) in comparison to the 50 km south Lake Sarez station (3290 m a.s.l), where average precipitation is only about 110 mm $a^{-1}$. Large glaciers (e.g. Fedchenko) and rock glaciers are abundant in the Western Pamir. The study site #457 is located within this sub-region. Geophysical profiles are located on a rock glacier and on fine-grained, mostly vegetated sediment at a mean altitude of 4580 m a.s.l. The closest meteorological station is located in the Jelondy village, about 30 km away from the site. Precipitation is not measured at this station. MAAT in the years 2020 - 2022 at an altitude of 3560 m a.s.l was -1.11 °C.

## 2.5 Eastern Pamir

The Westerlies in combination with the Indian Summer Monsoon (ISM) are the main drivers of the climate in the Eastern Pamir (Aizen et al., 2009). There is a negative west-east precipitation gradient (Fuchs et al., 2013; Pohl et al., 2015), making the Eastern Pamir the driest of all the regions investigated in this study, with very little overall precipitation (40 - 140 mm $a^{-1}$) (?Pohl et al., 2015; Barandun and Pohl, 2023). Most of the Eastern Pamir are further characterized by a high plateau (mean of 4000 m. a.s.l) with wide valleys. While large rock glaciers are present in the Southeastern Pamir, towards the Northeastern Pamir they diminish in size and frequency. The study sites Bulunkul, Zulmart, and Karakul are located in this sub-region, distributed along a North-South axis. Bulunkul in general is a special site due to geographic barriers surrounding the site, resulting in low precipitation and exceptionally low temperatures (Pohl et al., 2015). The geophysical profile (at an altitude of 3720 m a.s.l.) is located on fine-grained, partly vegetated sediment. The profiles at the Zulmart study site are located on rock glaciers, fine-grained sediment and on a moraine, all at a similar altitude of about 4530 m. a.s.l., with mean temperatures of about -4.1 °C (measured since 2021). The Karakul study site is located to the north of Lake Karakul at an altitude of 4200 m a.s.l. Measurements (1934 - 2017) from the meteorological station in the Karakul village, located 20 km to the south of the study site at an altitude of 3900 m a.s.l. indicate MAAT of -3.7 °C and low precipitation of about 80 mm a$^{-1}$. The geophysical profiles are located on fine-grained sediments.



# 3 Methods

We use two well-established geophysical techniques, ERT and RST, and a modified PJI approach to assess the permafrost distribution and the ground ice contents at the different study sites. The steps of our methodology are summarised in Figure 3 and will be introduced in the following sections.

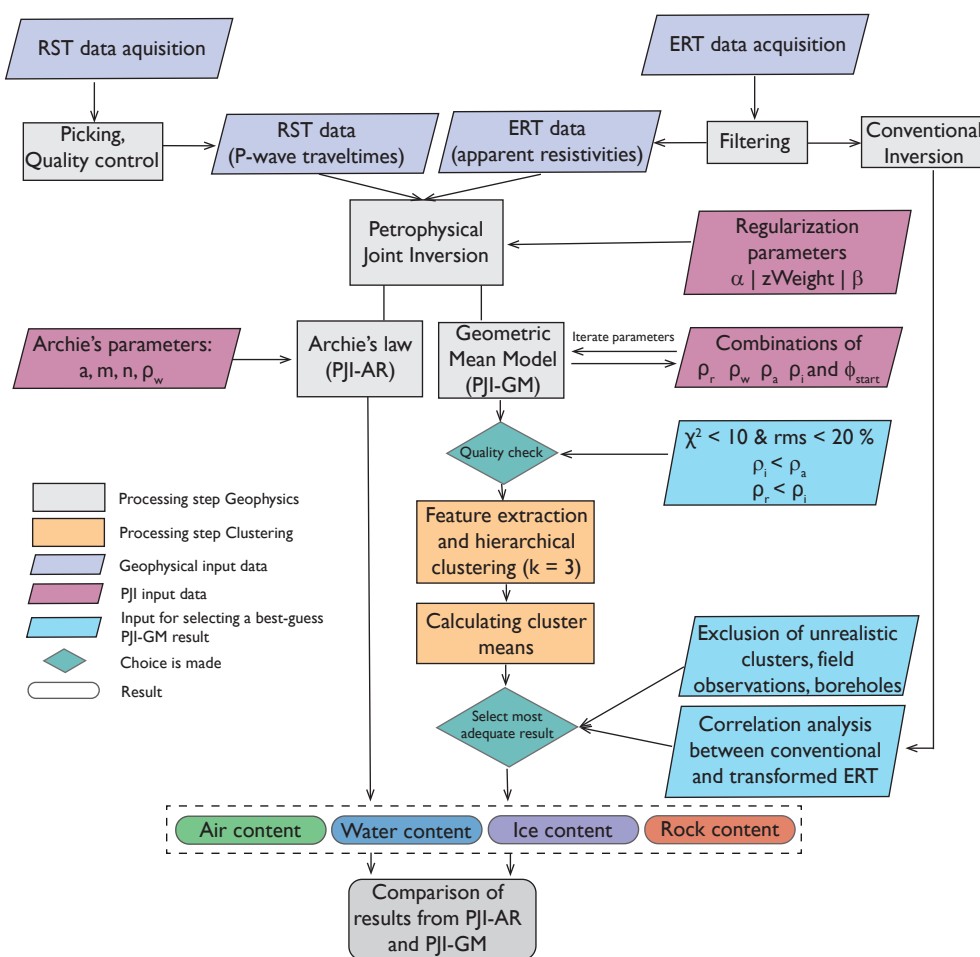

**Figure 3.** Methods flowchart: The flowchart shows the different steps of the methodological approach presented in this paper to estimate air, water, ice and rock contents of different permafrost landforms.

## 3.1 Electrical Resistivity Tomography (ERT) and Refraction Seismic Tomography (RST)

The ERT data were collected using a Syscal instrument (Iris instruments) with electrode spacing ranging from 2 m to 5 m, primarily employing the Wenner configuration for its good signal-to-noise ratio while also carrying out supplementary dipole-dipole measurements for enhanced lateral resolution at some of the profiles (see Table 2). To efficiently compare the gathered



data sets from diverse locations and landforms, it is essential to ensure consistent processing and analysis of the data (Herring
et al., 2023; Mollaret et al., 2019). Therefore, the measured apparent resistivity data were filtered with criteria proposed by
Mollaret et al. (2019) to eliminate outliers and to avoid overfitting of random noise contained in the data. Despite partly
challenging terrain conditions, more than 90 % of the original quadrupoles remained after this filtering approach, indicating
good overall quality of the raw data. A summary of the filtering statistics is presented in Table A1. The filtering is followed
by iterative data inversion using the open-source pyGIMLi library with a smoothness-constrained least-squares generalized
Gauss–Newton algorithm (Rücker et al., 2017). Regularization parameters for individual inversions include the smoothness
regularization parameter ($\alpha$), and a parameter for the relative weight for vertical boundaries (zWeight). These parameters
were individually selected for all profiles through a sensitivity analysis and using the L-curve method proposed in Rücker et al.
(2017) to optimize model response. The evaluation of inversion quality involved assessing the dimensionless error-weighted $\chi^2$
parameter, which quantifies the misfit between the model response and the data for a given data error, along with the RMS error,
which provides a measure of average magnitude of errors between observed data and model response. The hereby optimised
regularization parameters $\alpha$ and zWeight used in the individually inverted ERT data are in the following kept consistent with
those used in the PJI joint inversion runs.

Co-located on 22 of the 38 ERT profiles we conducted Refraction Seismics Tomography (RST) surveys using a Geode
system equipped with 24 geophones, also spaced 2 to 5 meters apart. RST first-arrival picking was performed using the
software ReflexW (Sandmeier K., 2024). Picking the first arrivals for each geophone can be challenging when dealing with
poor data quality, which may result from inadequate anchorage of the geophones in the ground or other disruptive factors such
as strong winds causing noise. Nevertheless, the correct identification of first-arrival travel times is a critical step in RST data
quality. To ensure good quality, only datasets with over 80 % confidently identified first-arrivals were considered suitable for
further processing, and subsequent data inversion to ensure reliable results. Only one RST dataset had to be excluded from the
further processing steps because of bad data quality. Similar to ERT, data inversion was carried out within pyGIMLi, where the
regularization parameters were chosen in the same way as for the ERT inversions, explained above. The quality of the inversion
results was assessed by forward-modeling of the ray paths and subsequent comparison with measured travel times. A summary
of the RST filtering and inversion statistics can be found in Table B1. It has to be noted that RST surveys take longer to conduct
compared to ERT surveys, so they were performed only at specific ERT profiles as shown in Fig. 2, and typically do not cover
the entire length of the ERT profiles.

## 3.2 Petrophysical Joint Inversion

To quantify ground ice content, we employ the Petrophysical Joint Inversion (PJI) approach, as developed by Wagner et al.
(2019). The PJI model combines a set of petrophysical equations to quantify subsurface ice, water, and air content based
on measured seismic P-wave travel times and electrical resistivities. While porosity is a necessary input in the original 4PM
formulation Hauck et al. (2011), it is often poorly known. Furthermore, utilizing independently inverted seismic and electrical
data can yield non-physical results (Wagner et al., 2019; Mollaret et al., 2020). The PJI model offers the advantage of a
simultaneous and physically consistent inversion for rock, ice, water, and air contents by using the apparent resistivities and





travel times as input data. The underlying assumption in the model is that the subsurface is composed of four phases: rock matrix ($f_r$), water ($f_w$), air ($f_a$), and ice ($f_i$):

$$f_w + f_r + f_i + f_a = 1 \tag{1}$$

We employ Equation 2, known as the time-average equation, to estimate the bulk velocity $v$ based on the constituent fractions and their respective velocities. This method is an expansion of Timur's time-averaging approach (Timur, 1968) to accommodate all four phases found in permafrost (Hauck et al., 2011). Equation 2 expresses that the inverse of the P-wave velocity $1/v$ (slowness) in a mixture is equivalent to the combined slownesses of each component, weighted by their respective volumetric
fractions:

$$\frac{1}{v} = \frac{f_w}{v_w} + \frac{f_r}{v_r} + \frac{f_i}{v_i} + \frac{f_a}{v_a} \tag{2}$$

The phase velocities $v_w$ (= 1500 $m/s$), $v_i$ (= 3500 $m/s$), and $v_a$ (= 300 $m/s$) were considered constant and equal to
values well-established in literature (e.g. Hauck and Kneisel, 2008). However, $v_r$, is site-dependent and could be defined with laboratory tests, which are not available at the sites investigated in this study. Therefore, we also considered $v_r$ constant for all sites, with an average value of 5500 $m/s$.

### 3.2.1 Petrophysical equations for electrical resistivity

### 3.2.2 Archie's law

Archie's law (Archie, 1942) is the most commonly used petrophysical equation which relates the bulk resistivity $\rho$ to the pore water resistivity $\rho_w$, the porosity and the fraction of the pore space occupied by liquid water:

$$\rho = \rho_w (1 - f_r)^{-m} \left( \frac{f_w}{1 - f_r} \right)^{-n}$$

where   $\rho_w$ is resistivity of pore water,

  $n$ is the saturation exponent,   (3)

  $m$ is the cementation exponent,

The exponents m and n are substrate-specific parameters assumed constant over space in our study, as no detailed subsurface information is available, which is generally the case for the study sites in Central Asia. However, Archie's law is considered



valid only when electrolytic conduction dominates. This is not universally justified for different substrates and landforms in mountainous terrain, especially for coarse-blocky material (Duvillard et al., 2018; Coperey et al., 2019). Archie's law relies on the assumption that electrical conduction occurs through the water in the pore space, which can lead to an underestimation of porosity when non-conductive phases, such as ice or air, are dominant. Also, other conduction mechanisms such as surface conduction in an electrical double layer or at the interface between the ice and the unfrozen material are neglected (Maierhofer et al. 2024). The advantages of Archie's law include its simplicity and long-established application in hydrogeology, but it can lead to misinterpretations of resistivity in mountain permafrost environments, often resulting in an overestimation of rock content due to its inability to account for non-conductive materials. Furthermore, the fractions of ice and air are not included in Archie's law (see Eq. 3), and are therefore not constrained by the equation. The PJI version using Archie's law is hereafter called PJI-AR.

### 3.2.3 Geometric Mean Model

The PJI model using the electrical Geometric Mean (PJI-GM) offers an alternative petrophysical equation to link the measured bulk resistivity to the volumetric fractions of the four phases. It has the advantage of including rock, ice and air resistivities in addition to the resistivity of the pore water resistivity ($\rho_{r,i,w,a}$ in Eq. 4) (Mollaret et al., 2020), similar to the P-wave velocity equation. It represents an alternative approach that might be better suited for environments with mixed conductive and non-conductive phases, such as permafrost environments with a mixture of rock, air, ice and water. This method assumes that the four phases are randomly distributed within the subsurface. (Somerton, 1992; Luo et al., 1994; Glover, 2010):

$$\rho = \rho_r^{f_r} \cdot \rho_i^{f_i} \cdot \rho_w^{f_w} \cdot \rho_a^{f_a} \tag{4}$$

However, the PJI-GM model (Eq. 4) encounters challenges due to the introduction of other unknowns, namely the resistivities of rock, ice, and air into the system of equations, which can be difficult to determine. Further, inclusion of $f_i$, $f_r$ and $f_a$ in Eq. 4 increases the coupling between the system of equations, which may potentially impact the inversion convergence, i.e. refers to the process by which the iterative algorithm reaches a stable solution that adequately fits the observed data. It can therefore result in larger misfits ($\chi^2$, and RMS errors) between measured and modeled data (Mollaret et al., 2020). The PJI-GM has not yet been tested extensively; here, we test the PJI-GM on the 22 profiles in Central Asia and assess its suitability for different landforms and substrates.

Following Hilbich et al. (2022), to facilitate comparisons of potential ground ice content across profiles, we define a Zone of Interest (ZOI) for each profile below the active layer from which a mean ground ice content is extracted. The manually delineated ZOIs for each profile are shown in the Annex (Figure A1).

### 3.2.4 Model setup and parameter choice

To determine the best-guess ground ice contents for individual profiles, a systematic analysis was conducted using the PJI-GM. This involved iteratively testing 450 different combinations of prescribed subsurface resistivity values ($\rho_r$, $\rho_w$, $\rho_i$, and $\rho_a$) and



start porosities ($\phi_{start}$). The start porosity (initially homogeneous across the profile) is a required initial value for the model and is iteratively adjusted during the inversion. The porosity is directly related to the rock content ($\phi$ = 1 - $f_r$). As our study areas

include very diverse landforms and substrates, the measured apparent resistivities exhibit significant variability across profiles and material-specific properties such as $\rho_r$, $\rho_w$ and $\rho_i$ are expected to vary as well (e.g. due to a different ion content in the soil and pore water). To address this variability, ranges of resistivity values representative for all landforms were chosen. For example, fine-grained sediments typically contain higher liquid water contents, leading to lower resistivities. Conversely, rock glaciers often display exceptionally high resistivities due to increased ice and/or air contents in the coarse blocky subsurface

matrix. The chosen resistivity ranges correspond hereby to physically plausible ranges found in the literature (e.g. Telford et al., 1990; Hauck and Kneisel, 2008). Table 3 indicates which parameter values were tested for $\rho_r$, $\rho_w$, and $\rho_i$. The resistivity of air ($\rho_a$) should theoretically be infinite. However, we found that for fine-grained sediment profiles, $\rho_a$ needed to be lowered significantly (i.e., set to 100'000 Ωm), otherwise the inversion would not converge at all and result in very high $\chi^2$ and RMS errors. As a result, we chose to treat ($\rho_a$) as a tuning parameter for model convergence, under the constraint that it always

remains higher than the resistivities of the other fractions.

**Table 3.** Values for $\rho_i$, $\rho_r$, and $\rho_w$ tested in the PJI-GM loop. All possible combinations were run except for physically implausible ones such as $\rho_r > \rho_i$ and/or $\rho_i > \rho_a$. Units are in Ohm meter (Ωm).

| $\rho_i$ (Ωm) | $\rho_r$ (Ωm) | $\rho_w$ (Ωm) |
|---|---|---|
| $5 \times 10^3$ | $1 \times 10^3$ | 2 |
| $1 \times 10^4$ | $2 \times 10^3$ | 10 |
| $5 \times 10^4$ | $5 \times 10^3$ | 20 |
| $1 \times 10^5$ | $1 \times 10^4$ | 100 |
| $5 \times 10^5$ | $2 \times 10^4$ | 150 |
| $8 \times 10^5$ | $3 \times 10^4$ | |
| $1 \times 10^6$ | $5 \times 10^4$ | |
| $2 \times 10^6$ | $1 \times 10^5$ | |
| $5 \times 10^6$ | | |

In addition to the varying resistivity values for the different materials, we tested different regularization parameters for the PJI inversions and chose the most adequate parameter values for each profile individually using the L-curve method (following Mollaret et al. (2020), Pavoni et al. (2023), Wagner et al. (2019), (Rücker et al., 2017)). In addition to the regularization parameters introduced earlier ($\alpha$, and zWeight), the PJI includes a volumetric conservation regularization parameter ($\beta$). This

was set to the default of 10'000 as we found that this value typically leads to satisfactory results regarding mass conservation.

Furthermore, we set the minimal rock content to 0.1 and the maximum rock content to 0.9 for all profiles to allow for the detection of bedrock (high rock content) and excess ice (very low to negligible rock content, e.g. in rock glaciers and ice wedges). This range was not restricted further to assess the model performance in an unbiased way and to see how the model



**Table 4.** Selected PJI model regularization parameters for each profile. Parameters m and n are only used in PJI-AR (Archie's law parameters). All other parameters are needed for both PJI-AR and PJI-GM. $err_{\rho_a}$ and $err_{tt}$ are data errors that were estimated for each profile iteratively to get $\chi^2$ values closer to 1.

| Profile | Landform | $\alpha$ | zWeight | $\beta$ | $\phi_{start}$ | m | n | $err_{\rho_a}$ (%) | $err_{tt}$ (ms) |
|---------|----------|----------|---------|---------|----------------|---|---|--------------------|------------------|
| abra02 | RG | 10 | 0.9 | 10'000 | 0.5 | 1.4 | 2.4 | 8 | 1.7 |
| abra03 | RG | 10 | 0.4 | 10'000 | 0.5 | 1.4 | 2.4 | 9 | 0.9 |
| abra04 | RG | 10 | 0.2 | 10'000 | 0.5 | 1.4 | 2.4 | 5 | 0.5 |
| abra05 | TS | 10 | 0.2 | 10'000 | 0.3 | 1.4 | 2.4 | 5 | 0.9 |
| abra07 | SED | 5 | 0.9 | 10'000 | 0.25 | 3 | 1 | 4 | 0.6 |
| abra09 | RG | 15 | 0.25 | 10'000 | 0.5 | 1.4 | 2.4 | 9 | 0.9 |
| abra10 | MO | 20 | 0.2 | 10'000 | 0.3 | 1.4 | 2.4 | 8 | 0.6 |
| BUL01 | SED | 20 | 0.1 | 10'000 | 0.25 | 3 | 1 | 6 | 1.2 |
| GOL01 | TS | 5 | 0.2 | 10'000 | 0.3 | 1.4 | 2.4 | 6 | 0.9 |
| GOL07a | SED | 10 | 0.5 | 10'000 | 0.25 | 3 | 1 | 8 | 1.4 |
| GOL07b | MO | 10 | 0.5 | 10'000 | 0.3 | 1.4 | 2.4 | 6 | 1.5 |
| GOL07c | RG | 5 | 0.9 | 10'000 | 0.5 | 2.4 | 3.1 | 6 | 1.5 |
| no457 | SED | 10 | 0.1 | 10'000 | 0.25 | 3 | 1 | 9 | 1.0 |
| SUE01 | TS | 30 | 0.2 | 10'000 | 0.3 | 1.4 | 2.4 | 5 | 1.0 |
| SUE02 | TS | 10 | 0.9 | 10'000 | 0.3 | 1.4 | 2.4 | 8 | 0.6 |
| SUE05 | MO | 10 | 1 | 10'000 | 0.3 | 1.4 | 2.4 | 6 | 0.8 |
| KAR01 | SED | 10 | 0.5 | 10'000 | 0.25 | 3 | 1 | 6 | 0.6 |
| KUM01 | SED | 5 | 0.3 | 10'000 | 0.25 | 3 | 1 | 9 | 0.1 |
| KUM02 | SED | 20 | 0.1 | 10'000 | 0.25 | 3 | 1 | 8 | 1.0 |
| KUM04 | SED | 25 | 0.1 | 10'000 | 0.25 | 3 | 1 | 3 | 1.0 |
| ZUL01 | RG | 20 | 0.2 | 10'000 | 0.5 | 1.4 | 2.4 | 3 | 0.5 |

performs if the porosity is unknown, which is often the case for remote regions without boreholes. The PJI model parameters
used for each profile can be found in Table A1.

### 3.3 Clustering approach for tomogram analysis

After running the large number of PJI-GM simulations with the different parameter combinations indicated above, a best-guess
representation of the subsurface has to be chosen. However, even after removing all models with a large misfit identified in
a quality check (Fig. 3, model runs where $\chi^2$ and RMSE were above threshold), the number of PJI-GM model runs with
325 satisfactory model fits was still large for many profiles. This is a result of the non-uniqueness of the inversion process, wherein
multiple subsurface models can effectively represent the same geophysical data. To assess the range of possible subsurface





conditions modeled with PJI-GM, we conducted a hierarchical cluster analysis (analogous to the ensemble inversion approach by Rings and Hauck (2009)). This analysis helps to identify and group similar inversion results based on the mean ground ice content and its standard deviation. The clustering steps can be summarized as follows:

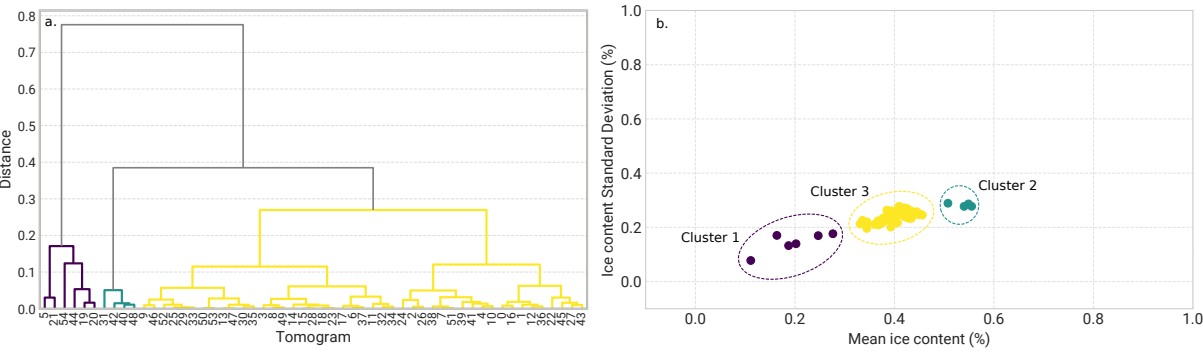

**Figure 4.** Dendrogram and scatterplot illustrating the hierarchical clustering of PJI-GM model outputs for the abra02 rock glacier profile (here, for the 53 remaining tomograms after the quality check). (a) Dendrogram resulting from the hierarchical clustering of the extracted features. (b) Scatterplot of mean ice content and the ice content standard deviation with points colored according to their respective clusters, demonstrating the differences in mean ice content and variability among the clusters.

1. Feature Extraction: From each remaining tomogram realisation after the initial quality check (Fig. 3), key features (mean and standard deviation) of ice content distribution were extracted. This feature extraction step facilitated the comparison of the tomograms and grouping into similar clusters (see Figure 4 for an example).

2. Hierarchical Clustering: Subsequently, to analyze similarities across all tomograms and identify overarching patterns, a hierarchical clustering was conducted grouping all tomograms of each profile into three clusters which represent different types of modeled ice content distributions. We performed hierarchical clustering using the Ward's method to group the tomograms based on their extracted features. The clustering analysis was conducted using the scipy.cluster.hierarchy module from the SciPy library (version 1.11.2, Virtanen et al. (2020)). We chose the same number of clusters (3) for each profile for easier comparison across profiles. The dendrogram plot (see Figure 4a) visually represents the hierarchical arrangement of clusters, where each individual tomogram is represented at the bottom. The height of the branches indicates the distance or dissimilarity between clusters, reflecting how much variance increases when merging clusters. Clusters merging at smaller distance are more similar than those merging at larger distance, providing insights into the similarity and structure of the tomograms.

3. Finally, the ice, rock, water and air contents from all tomograms within the same cluster are averaged to obtain a single representative result for the four fractions for each cluster. This results in three representative tomograms, corresponding to the three identified clusters, that show different possibilities of the phase distributions in the subsurface.





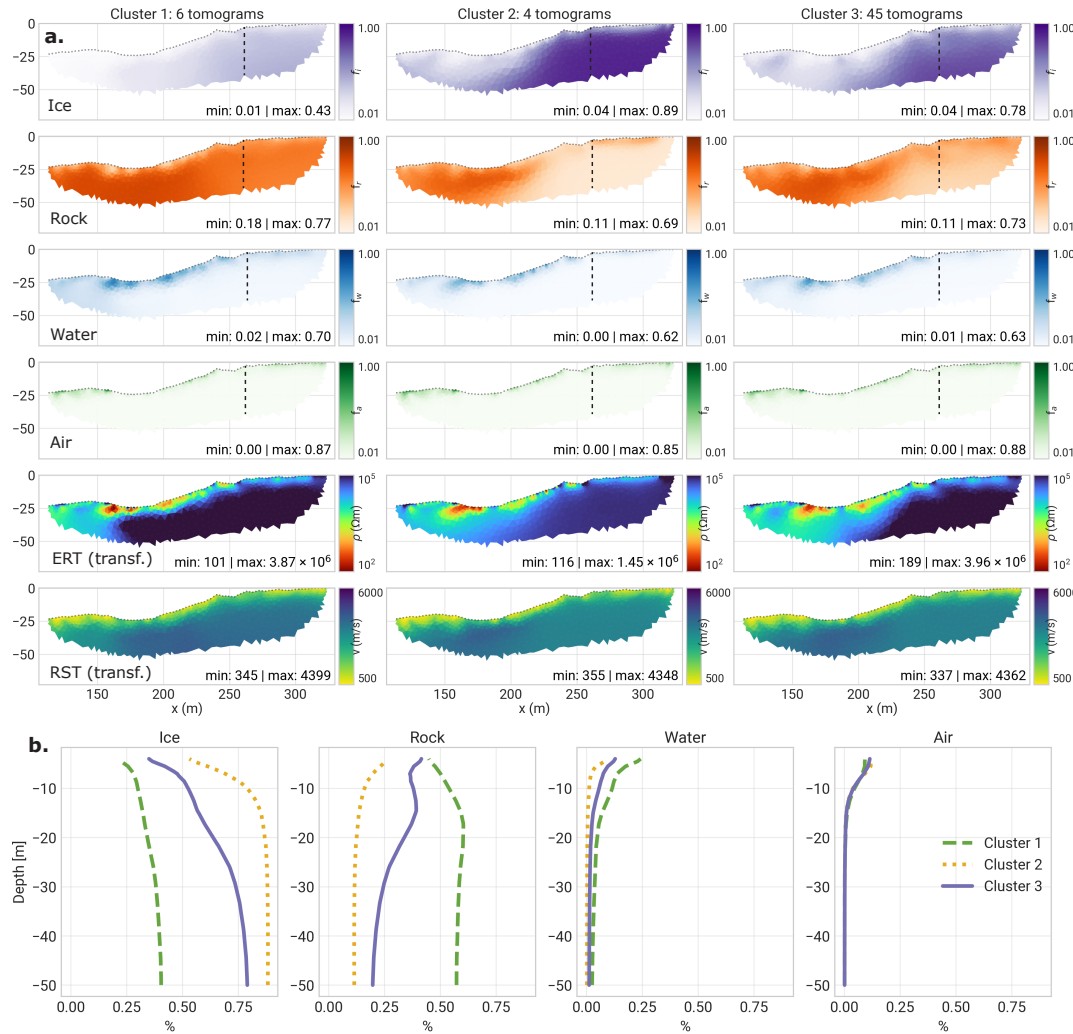

**Figure 5.** Example of the PJI-GM clustering results for profile abra02 on Abramov rock glacier. (a) PJI-GM results (ice, rock, water, air contents) for each cluster, as well as the transformed resistivity (ERT) and P-wave velocity (RST) distributions calculated from these distributions. (b) Virtual borehole plots which, for easier comparison, represent the four fractions of all PJI-GM clusters at one point (x = 260 m) with depth.

The clustering approach allows for comparing the variability in ground ice content distribution across multiple model runs. Grouping similar tomograms into clusters representing different subsurface models based on the geophysical data facilitates analyzing the large number of PJI-GM model outputs. Figure 5 exemplifies this with three identified clusters for profile abra02, showcasing their tomographic visualizations (ice, rock, water, and air fractions, along with ERT and RST tomograms). Additionally, Figure 5b provides a virtual borehole plot at x = 260 m, illustrating the depth profiles of these subsurface fractions. To determine the best-guess ground ice content estimate from these PJI-GM clusters, we implemented a multi-step approach.



Initially, unrealistic subsurface characteristics were identified and flagged for exclusion. These characteristics, with their corresponding rejection thresholds, are detailed in Table 5.

– Abrupt, unrealistic transitions: Sudden and unrealistic shifts in the vertical distribution of any of the four subsurface components (rock, ice, water, air) or transformed ERT or RST tomograms that lacked a clear justification based on the site conditions.

– Implausible air content: Air content exceeding reasonable limits for the given soil type and depth. For instance, high air content at depth within fine-grained sediments, where compaction and saturation are expected, would be flagged as unrealistic.

– Implausible water content: Excessively high water content at depths where drainage would likely occur was also considered unrealistic.

– Implausible low rock content at the surface and at depth of fine-grained sediments.

Figure A provides an overview of this workflow, including notes on which cluster of each profile was rejected and the rationale for those rejections.

**Table 5.** Threshold ranges for cluster rejection based on unrealistic subsurface fractions at different depths. The top 5 m were excluded from this analysis, which in most cases excludes the active layer. CB = coarse-blocky (RGs, some MOs); FG = fine-grained (SED, TS)

| Depth | Sediment | Air (%) | Water (%) | Rock (%) | Rationale for Rejection |
|-------|----------|---------|-----------|----------|-------------------------|
| 5-10 m | CB | >30 | >20 | - | Air content decreases with depth but remains higher than in fine-grained sediments; limited water retention. |
| | FG | >20 | >35 | <25 | Compaction reduces air and water content, but water retention is typically higher than in coarse-blocky sediments. Rock content >25 % assumed for all fine-grained sediment profiles in our study sites. |
| > 10 m | CB | >5 | >20 | - | Increased compaction, thus limited air and water contents. |
| | FG | >5 | >30 | <60 | Significant compaction minimizes air and water content. |

This assessment was further validated by site-specific geomorphological observations and, where available, borehole data. For instance, the presence of landforms indicative of high ice content (e.g., active rock glaciers) or other indicators for ground ice presence (e.g., gelifluction lobes, thermokarst, visible ice outcrops) were considered. While borehole data provided valuable validation at the Kumtor site, such data were not available for the other sites. When multiple plausible clusters remained, we conducted a correlation analysis, comparing the transformed ERT from each PJI-GM cluster with the conventional ERT inversion. The cluster with the highest correlation, indicating the best fit to the measured data, was selected (Figure 6). This





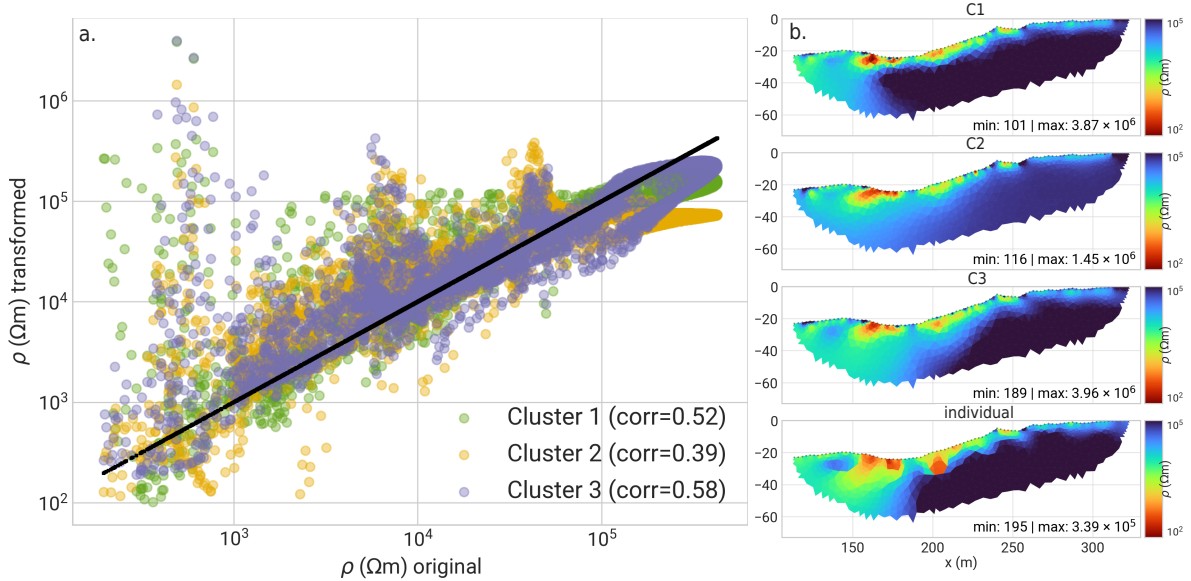

**Figure 6.** (a) Correlation plot showing the relationship between the inverted resistivity values obtained from individual inversion of the original ERT (ERT-conv) measurements and the transformed resistivity (ERT-transf) values from the different PJI-GM clusters (mean value of all tomograms of one cluster). The term "transformed" refers to the back-calculation of resistivity from the inversion results produced by the PJI-GM model. In this example (profile abra02), Cluster 2 can be excluded as it has the poorest fit. (b) Transformed ERT tomograms of all clusters (C1, C2, C3) versus the tomogram resulting from the individual inversion (chapter 3.1)

ensured the chosen model best reflected the actual subsurface conditions captured by the ERT measurements. Transformed RST tomograms were not used due to their similarity across clusters. We acknowledge that scarcity of validation data in the region, with borehole data limited to the Kumtor site, necessitates a degree of expert judgment in this selection process.

# 4 Results

## 4.1 Permafrost characteristics of the different sites and landforms in Central Asia

We collected 38 ERT measurements from 10 distinct locations to assess permafrost presence and qualitatively estimate ground ice content at each profile, distinguishing between ice-rich and ice-poor conditions. Additionally, 22 of these profiles were co-located with RST measurements, where the PJI method was applied. Our initial examination focused on identifying potential patterns in resistivity across the study sites and landform classes, with the aim of generalizing resistivity signatures by landform. Figure 7 shows the resistivity signature of each profile, grouped by the different landforms. The violin plots represent the full range of resistivity values present within each profile and landform, visualizing both the distribution and density of these values. The width of each violin at any given point reflects the relative frequency of that resistivity value, with wider sections indicating more common values and narrower sections highlighting less frequent occurrences. These plots emphasize





the variation in resistivity within different landforms, with distinct patterns emerging for each. For example, rock glaciers

(B) exhibit higher resistivity values, often associated with ice-rich subsurface conditions, whereas fine-grained sediments (C) display lower resistivity ranges, typical of more water-saturated or fine-grained materials. Talus slopes (D) and moraines (A) show intermediate resistivity distributions, potentially reflecting a mixture of rock, ice, and potentially unfrozen materials. Rock glaciers exhibited the highest resistivities (mean = 700'000 Ωm), followed by moraines (mean = 95'000 Ωm), indicating the presence of the ice in the core of the moraine.

The lowest resistivities were recorded for the fine-grained sediment sites (mean = 5500 Ωm). Talus slope resistivities (mean = 25'000 Ωm) fell between fine-grained sediments and moraines. The resistivity distributions across the different landforms show no significant variation between the geographic regions studied. These results suggest that resistivity signatures from the ERT surveys appear to be landform-specific rather than site or region-specific. However, the resistivity ranges within one profile are generally quite large, as indicated by the large vertical spread of the violin plots in Fig. 7.

To further understand permafrost occurrences and conditions across the different landforms in Central Asia, all ERT profiles were independently interpreted with a focus on the likelihood of permafrost presence and potential ground ice content. Layers of high resistivity, typically located beneath a lower resistivity active layer, were indicative of permafrost conditions. The dataset suggests widespread permafrost across all sites and landform types studied. In rock glaciers and moraine surveys particularly high resistivities were observed indicating ice-rich conditions (Fig 8 abc). Notably, exceptionally high resistivities

in moraine samples suggest the presence of buried glacier ice in many moraines within the Central Asia study sites, as seen in moraine profiles such as GOL05 (Fig. 8b).

In addition to the more commonly studied permafrost landforms such as rock glaciers and talus slopes, we also conducted measurements on numerous fine-grained, partially vegetated sediment profiles that are prevalent on the high-altitude mountain plateaus of Central Asia. A new borehole was drilled at the Kumtor study site in 2022 where profile KUM04 is located (see Fig.

8f), revealing frozen conditions until at least the maximum measurement depth of 32 m, a shallow active layer ranging from approximately 1 to 1.5 m thickness, and saturated ground ice conditions in the upper part of the drill core. Visual inspection of the core revealed fine-grained sediments with a notable presence of interstitial ice, although quantifying the exact ice content was difficult. Based on the sediment structure observed, we estimate a porosity of approximately 20-25 %, which would imply a maximum ice content of 25 % under saturated conditions, such as is the case in the uppermost few meters below the active

layer. At greater depths, while the borehole temperatures remain < 0 °C, the sediments appear much more dry, indicating decreasing ice contents with depth. The low resistivity values (around 5,000 Ωm) in these fine-grained sediments suggest that either the ice content is relatively low (likely around 20 %, as indicated by our visual interpretation of the KUM04 drill core) and/or the sediments retain a significant amount of unfrozen water. Comparable resistivity values (Fig. 7c) were observed in other fine-grained sediment profiles at other sites, implying similar subsurface conditions in those areas.

## 4.2 PJI-GM clustering results

We used a clustering approach to evaluate the PJI-GM and select a best-guess cluster. Figure 9 uses virtual borehole plots to illustrate the variability in subsurface composition (ice, rock, water, and air content) between different clusters for four





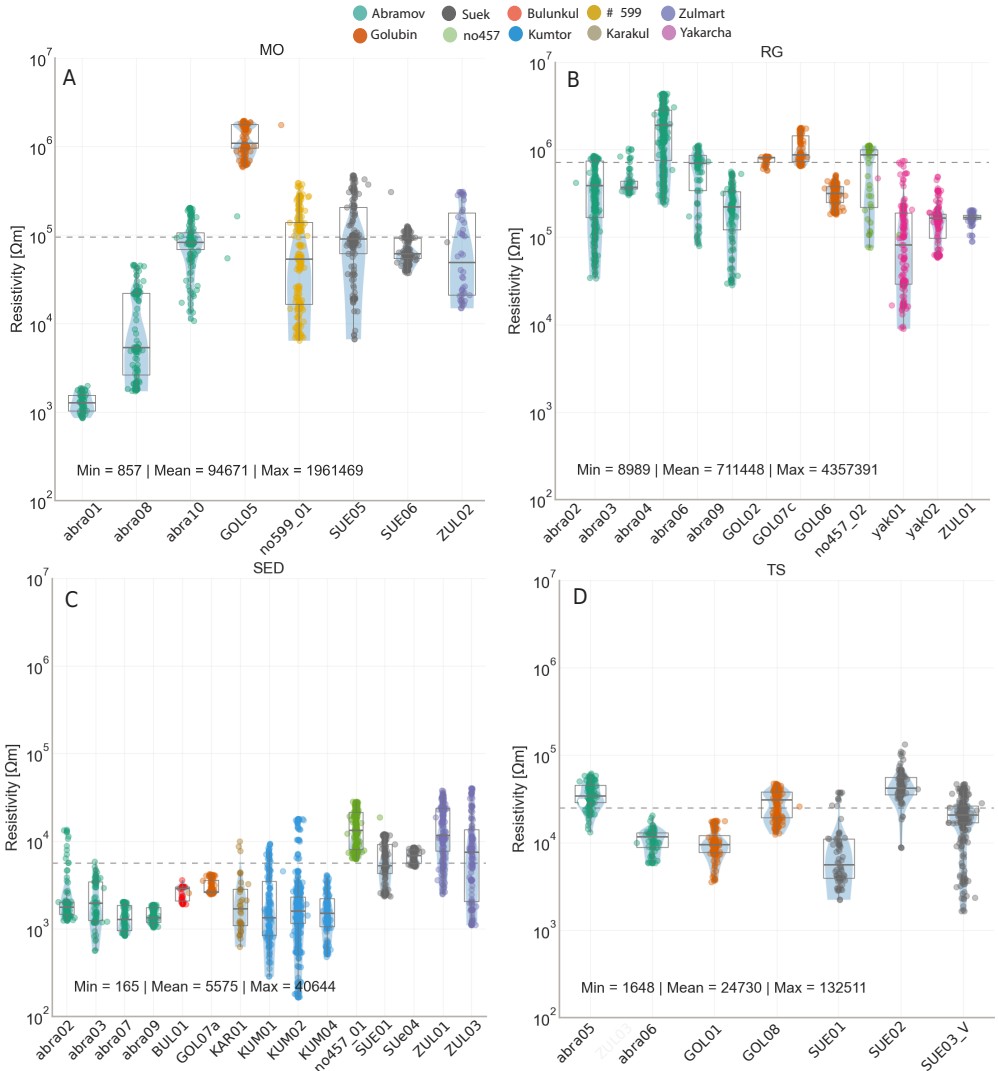

**Figure 7.** Resistivity distribution of all ERT profiles, grouped by landforms, and colored by study site. The horizontal dashed line in each subplot indicates the mean resistivity of all profiles within the landform class. (A) Moraine, (B) Rock Glacier, (C) Fine-Grained Sediment, (D) Talus Slope. Each point represents individual resistivity value from the inverted resistivity distribution, capturing the full range of resistivity values obtained across the entire profile. The wide sections of each violin indicate more frequent resistivity values, while narrower sections reflect less common resistivity values. This illustrates the diversity of resistivity measurements within each profile, highlighting variations that occur at different depths and positions along the landform.

representative profiles and landforms. Across most profiles, ice and rock content showed the greatest variation between clusters, while water content remained relatively consistent. Air content, consistently low across clusters, appears well-constrained by the model. However, in some profiles, the PJI-GM overestimated surface water content and underestimated rock content,





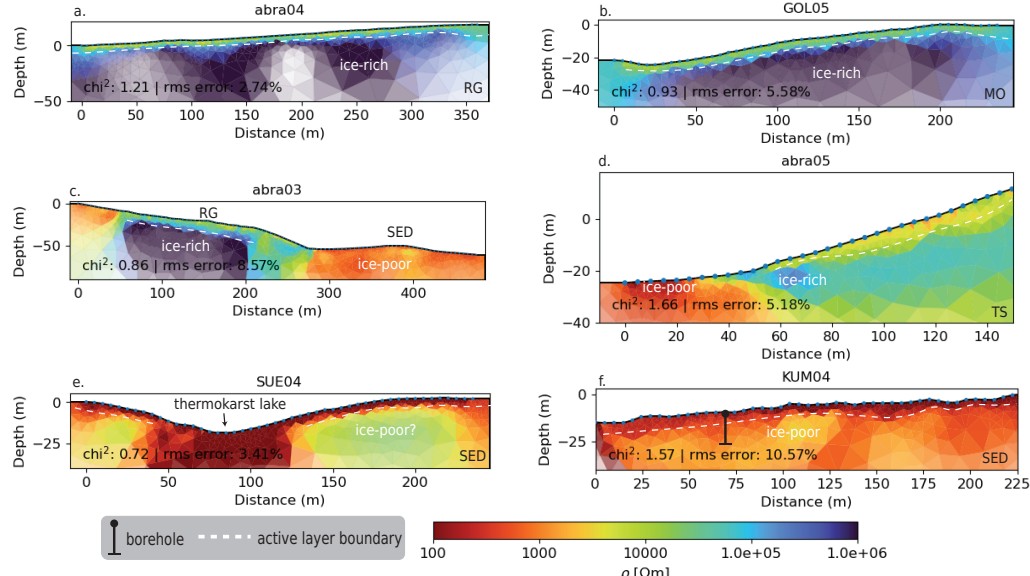

**Figure 8.** Examples of interpreted ERT profiles of different landforms. (a) rock glacier with high resistivities below an active layer of about 4 m; (b) moraine, high resistivities may point to buried glacier ice; (c) Rock glacier and fine grained sediments (d) talus slope; (e) fine-grained sediments; (f) fine-grained sediments, where a borehole confirmed saturated ground ice conditions in uppermost layers.

resulting in unrealistic water content values exceeding 50 % (Fig. 9c). This discrepancy was particularly pronounced in fine-grained sediment profiles, where higher surface rock content is expected than what was modeled for some clusters.

To quantify ground ice content, we extracted the mean ice content within a representative zone below the active layer (ZOI, Fig. A2) for each profile and PJI-GM cluster (Fig. 10). This is illustrated in Figure 10, where the selected best-guess PJI-GM cluster for each profile is highlighted with a red outline. Clusters that were rejected based on unrealistic results in any of the tomograms are marked with pink crosses. The figure also shows the mean ice content from the PJI-AR runs as a direct comparison. This will be further discussed in chapter 4.3. Furthermore, the ice content tomograms of all clusters for the four representative profiles/landforms are shown in Figure 11. In the following, we summarize some of the main results for each of the four landform classes, focusing on the ice contents.

- Moraines: The PJI-GM results for the three moraine profiles show considerable variability in ice content between individual clusters, highlighting the method's sensitivity to parameter choice. For instance, in the abra10 profile, mean ice content within the clusters ranged from 0 % to 55 % (Fig. 10a). The abra10 and SUE05 profiles appear to feature ice-rich permafrost conditions, as evidenced by the visible presence of ice outcrops within the associated moraine deposits. Conversely, lower ice contents were found in older, more distant moraines with finer-grained sediments, like GOL07b. The estimated mean ground ice content in the sampled moraines is between 15 % and 38 %.



**Figure 9.** Virtual borehole plots of all clusters and mean subsurface fractions of four representative profiles/landforms: (a) SUE05 (MO), (b) abra02 (RG), (c) KUM04 (SED), (d) abra05 (TS). Red arrows mark examples of unrealistic cluster results.





- Rock glaciers: Analysis of the six rock glacier profiles using PJI-GM yielded consistent results across clusters, with all clusters indicating the presence of ground ice (Fig. 10b). Best-guess clusters, selected through a combination of correlation analysis and field observations, yielded ground ice content estimates ranging from 40 % to 60 %, effectively reflecting the high ice contents expected in these landforms. The relatively low variability between clusters and the consistently realistic subsurface tomograms suggest well-constrained conditions for the PJI-GM model in the rock glacier profiles.

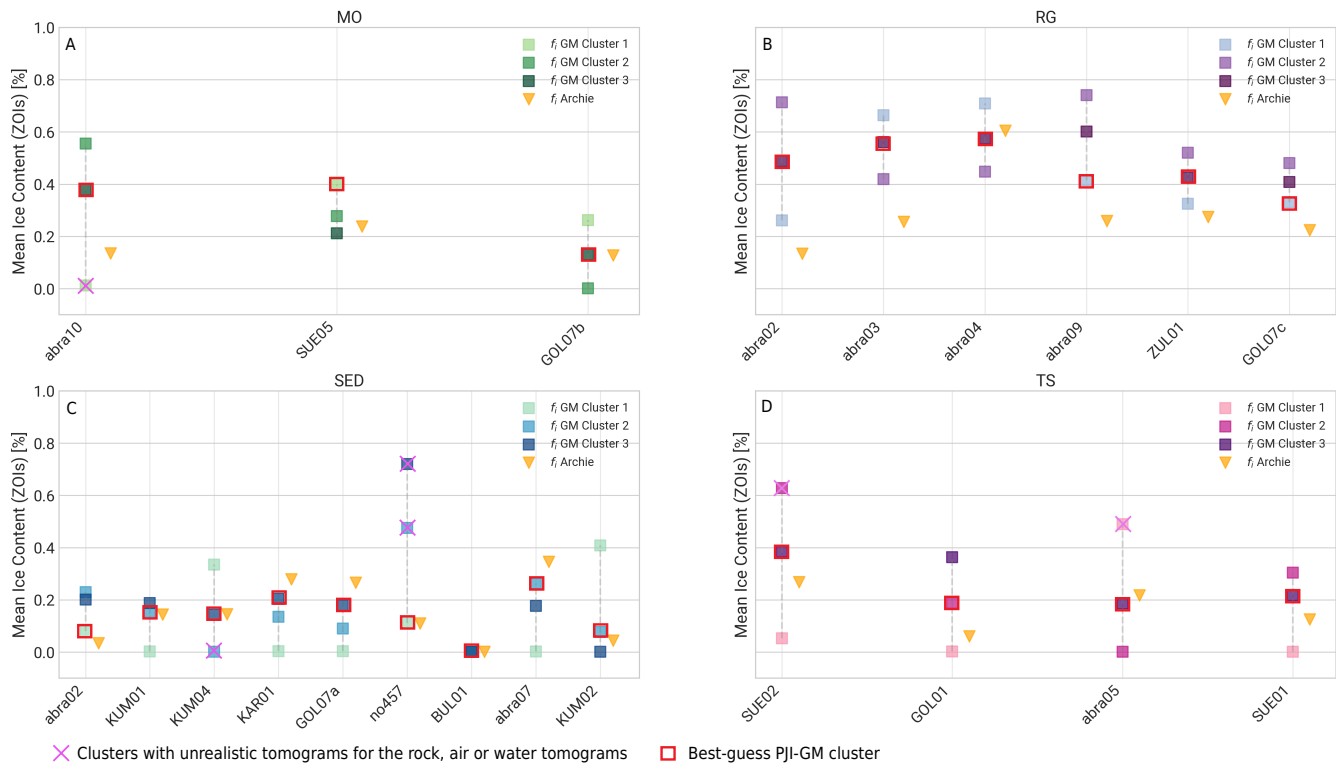

**Figure 10.** Mean ice content extracted from the zone of interest for each profile, comparing results from the three PJI-GM clusters with those obtained using Archie's law. The 'best guess' PJI-GM ground ice content result for each profile is emphasized with a red outline. Clusters that generate unrealistic tomograms for any of the other three subsurface fractions (rock, water, or air), or where field observations demonstrate that the cluster does not accurately represent the subsurface conditions, are marked with pink crosses and are therefore not considered valid representations of the subsurface. (A) Moraine, (B) Rock Glacier, (C) Talus Slope, (D) Fine-Grained Sediment.

- Fine-grained sediments: The PJI-GM model exhibited significant variability in ground ice content estimates between clusters for fine-grained sediment profiles. While most profiles yielded mean ground ice contents within the ZOI ranging from 0 % to 20 %, some exceptions exceeded this range, with profile no457 showing the greatest variability (10 % to 70 %). This inconsistency highlights the challenge of characterizing ground ice in these environments using PJI-GM. For example, in profile KUM04, where ice-saturated conditions were observed in the uppermost layers, the majority





of PJI-GM runs indicated a complete absence of ground ice (Fig. 11). Despite these challenges, best-guess estimates, informed by correlation analysis, borehole observations, and the exclusion of unrealistic tomograms, suggest a range of mean ground ice contents of 0 % to 23 %. However, systematic parameter estimation for this landform class using the current PJI-GM model remains difficult.

– Talus slopes: The PJI-GM model results for talus slope profiles also exhibit considerable variability. The differences in mean ice content between clusters range from 0 % to as high as 63 %. Some model clusters likely overestimated the ground ice content in talus slopes, leading to unrealistic rock fraction tomograms and their subsequent exclusion from the best-guess selection. The best-guess clusters were selected based on the highest correlation between conventional and transformed ERT data, with mean ground ice contents calculated for talus slopes ranging from 20 % to 40 %.

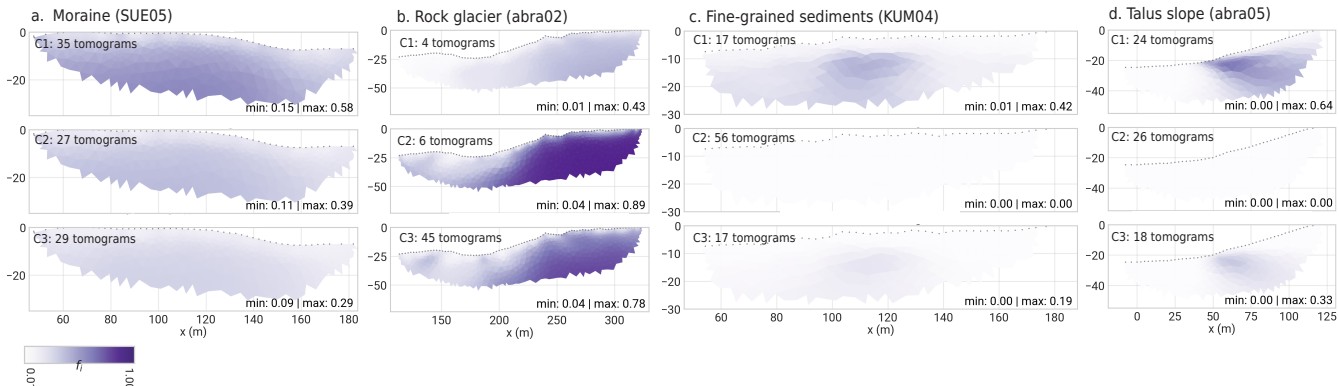

**Figure 11.** Comparison of the ground ice content results of the three PJI-GM clusters for a moraine (a), a rock glacier (b), fine-grained sediment (c) and a talus slope (d). For each cluster, the number in the top left corner indicates how many tomograms (or PJI-GM model runs) correspond to this cluster.

## 4.3 Comparison of Archie's Law (PJI-AR) and the Geometric Mean Model (PJI-GM)

We compared the best-guess PJI-GM cluster results with the more commonly used PJI-AR runs, maintaining consistent regularization parameters, to evaluate the suitability of each model version for the four landform classes. While ground truth data is limited to the Kumtor site, the PJI-GM model consistently produced tomograms with more distinct subsurface structures. In contrast, the PJI-AR model yielded more uniform subsurface fraction distributions across all landforms, as evidenced by lower standard deviations in the tomograms (see also ice content boxplots in Fig. A4). This improvement with the PJI-GM model is likely attributed to its ability to more precisely characterize the subsurface porosity distribution.

Compared to PJI-GM, PJI-AR consistently produces lower ground ice content estimates for most rock glacier profiles (Figure 10b), averaging around 25 %. This seems underestimated given previous findings on rock glacier ice content (e.g. Arenson et al., 2002; Monnier and Kinnard, 2013; Scapozza et al., 2015; Bast et al., 2024). The discrepancy likely arises from PJI-AR's



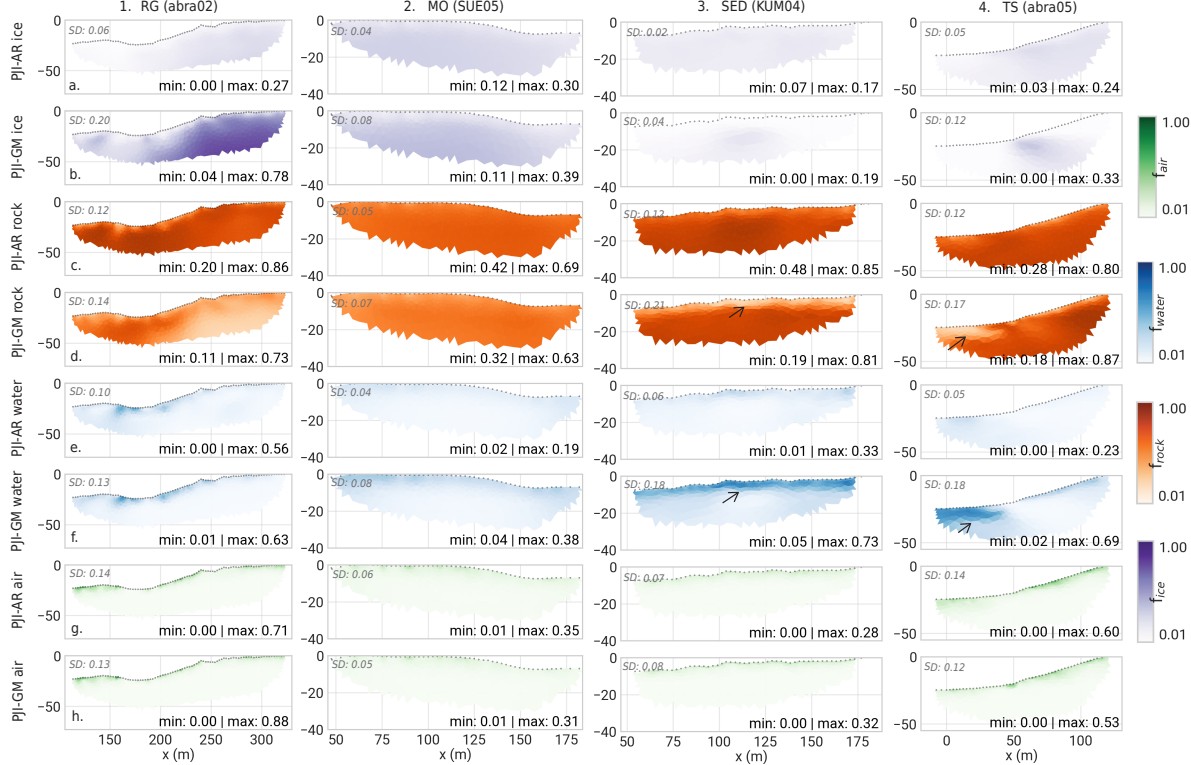

**Figure 12.** Comparison of PJI-AR and best-guess PJI-GM model results for all four fractions (ice, rock, water, air) of the four representative profiles introduced in Fig. 9. The arrows indicate the clear over-estimation of water contents and underestimation of rock contents in profiles abra05 (TS) and KUM04 (SED) from the PJI-GM model. The standard deviation of each fraction is noted on the top left corner of each tomogram.

difficulty in distinguishing between ice and rock in high-resistivity settings, which are characteristic of rock glaciers with pore spaces filled with non-conductive fluids. While PJI-AR performs comparably to PJI-GM for profile abra04, PJI-GM generally resolves this ice-rock ambiguity more effectively in all the other RG profiles, generating more realistic ice content estimates and delineating subsurface structures more clearly (Figure 13). The figure shows that the PJI-AR model yields spatially uniform results for both rock and ice fractions, while the PJI-GM model provides a clearer delineation of the active layer and other structures. This is also quantitatively supported by the higher standard deviations for the PJI-GM (see Fig. 121a vs. 1b). Notably, both models produce similar results for the other subsurface fractions of rock glaciers.

For talus slopes and moraines, the PJI-AR model generally provides more realistic ice content estimates compared to its performance for rock glaciers, with results often aligning well with those from the PJI-GM model (Figure 10c and d). However, both models exhibit limitations. In cases where field observations suggest more ice-rich conditions (abra10, SUE05), the PJI-AR model appears to underestimate ground ice contents for TS and MO profiles too. On the other hand, the PJI-GM model frequently overestimates water content in areas characterized by low resistivity, particularly in the near-surface layers of talus




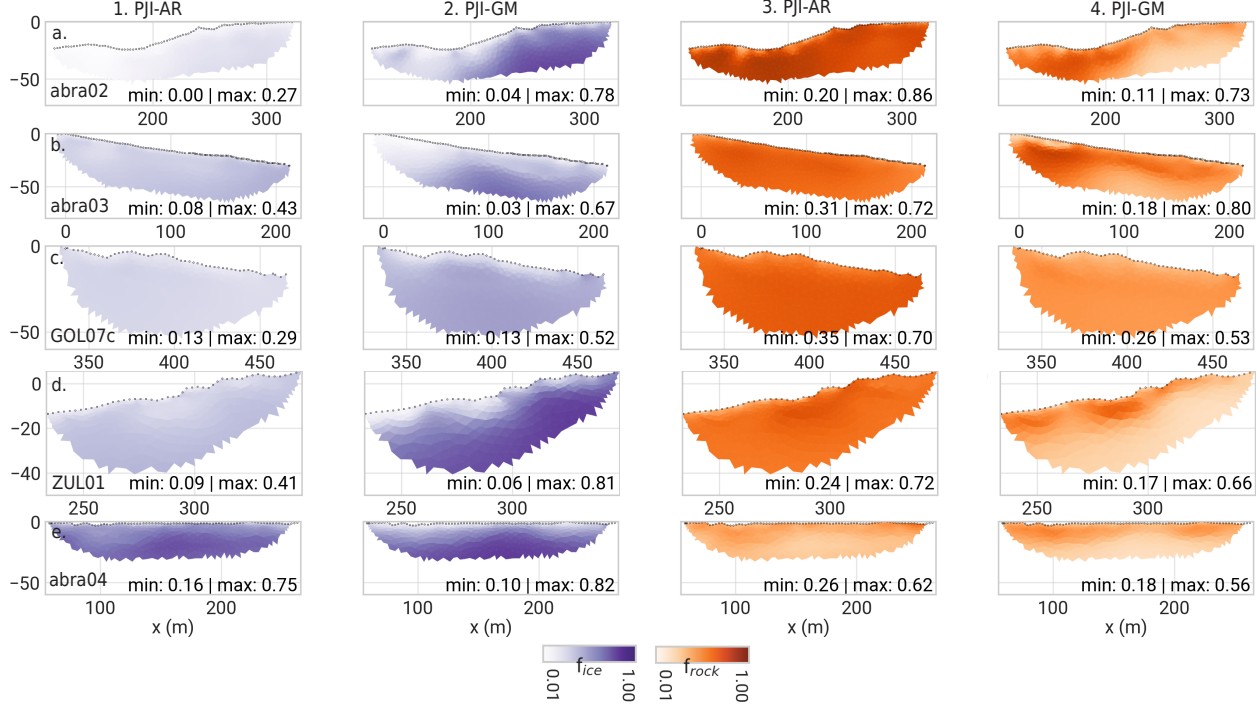

**Figure 13.** Comparison of ice and rock contents modeled with PJI-AR and PJI-GM for different rock glaciers. Columns 1 and 3 show the ice and rock content modeled with PJI-AR while columns 2 and 4 show the same but modeled with the PJI-GM.

slopes, moraines, and fine-grained sediment profiles (Figure 12 3f and 4f). This overestimation is evident in cases like profile abra05, where PJI-GM predicts water content exceeding 60 % at the surface, contradicting field observations. This tendency to overestimate water content often leads to unrealistic results and subsequent exclusion of PJI-GM clusters, especially in SED profiles. In contrast, PJI-AR does not exhibit this overestimation of water content. Otherwise, PJI-AR appears to be more suitable for characterizing ground ice content in fine-grained sediment profiles, particularly when extensive and time-consuming parameter optimization for PJI-GM is not feasible. This is supported by PJI-AR's accurate capture of low ice content in profiles KUM01, KUM02, and KUM04, where PJI-GM often resulted in tomograms with 0 % ice content or failed to converge.

## 4.4 Model sensitivity analysis

A sensitivity analysis of the PJI-GM model, using 450 different resistivity combinations ($\rho_a$, $\rho_w$, $\rho_r$, $\rho_i$) for each profile within the PJI-GM model, revealed that ice content estimates are most sensitive to the choice of $\rho_i$ and $\rho_r$. Increasing these resistivities generally decreased ice content estimates (Figure 14), except for most fine-grained sediment profiles (Figure 14c), where no correlation was found. The analysis further revealed that variations in the parameter values of the water resistivity ($\rho_w$) did not consistently influence the resulting ice content estimates significantly. Nonetheless, it was observed that lower $\rho_w$





values, ranging from 2 to 20 $\Omega$m, generally produced better model performance across all landforms compared to higher $\rho_w$ values, which less frequently led to model convergence (not shown). Notably, for fine-grained sediment profiles, the ice content consistently approached 0 % whenever $\rho_w$ exceeded 20 $\Omega$m.

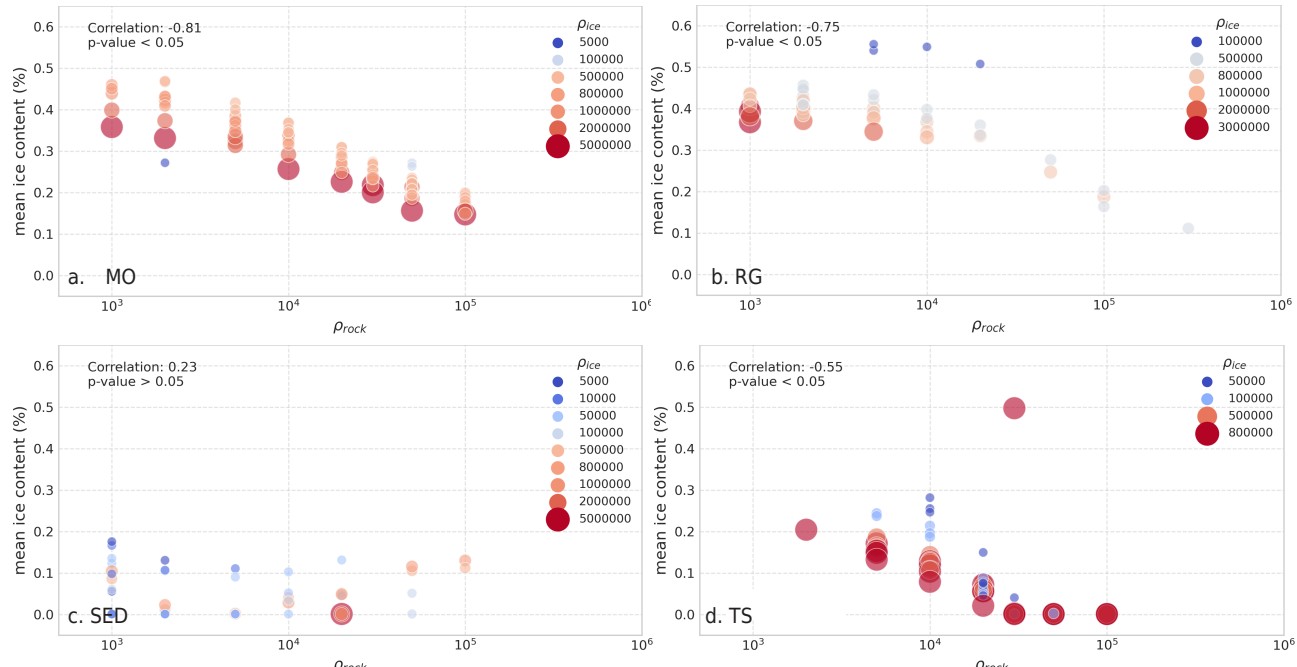

**Figure 14.** Influence of $\rho_r$ and $\rho_i$ on resulting ice content for example profiles of (a) moraine, (b) rock glacier, (c) fine-grained sediment and (d) talus slope.

We investigated the influence of start porosity on modeled ice contents, hypothesizing that higher initial porosities might lead to higher final porosities and thus, larger ice contents. To test this, we ran the PJI-GM model with varying start porosities (20-80 %) while keeping all other parameters constant, using median resistivity values from the best-guess clusters for four example profiles (Table 6). Figure 15 shows the mean values calculated across the entire tomograms for the four representative profiles. Contrary to our hypothesis, start porosity does not seem to significantly influence ice, water, rock, and air fractions in

most cases. For instance, rock glacier profile abra02 showed only a slight increase in mean ice content (35 % to 40 %) up to a start porosity of 50 %, but remaining constant thereafter. Similarly, profiles abra05 and SUE05 showed no clear increasing trend. However, the analysis highlighted the PJI-GM sensitivity to parameter changes. Profile KUM04 exhibited poor model fit ($\chi^2 > 10$) for most start porosities, indicating that even minor parameter adjustments can hinder convergence. This highlights the need for careful parameter selection and interpretation of model results.

Figure 15 also illustrates the impact of the start porosity on the PJI-AR model version, revealing that higher initial porosities result in increased mean ice contents, particularly for the fine-grained sediment profile (KUM04) and the rock glacier profile



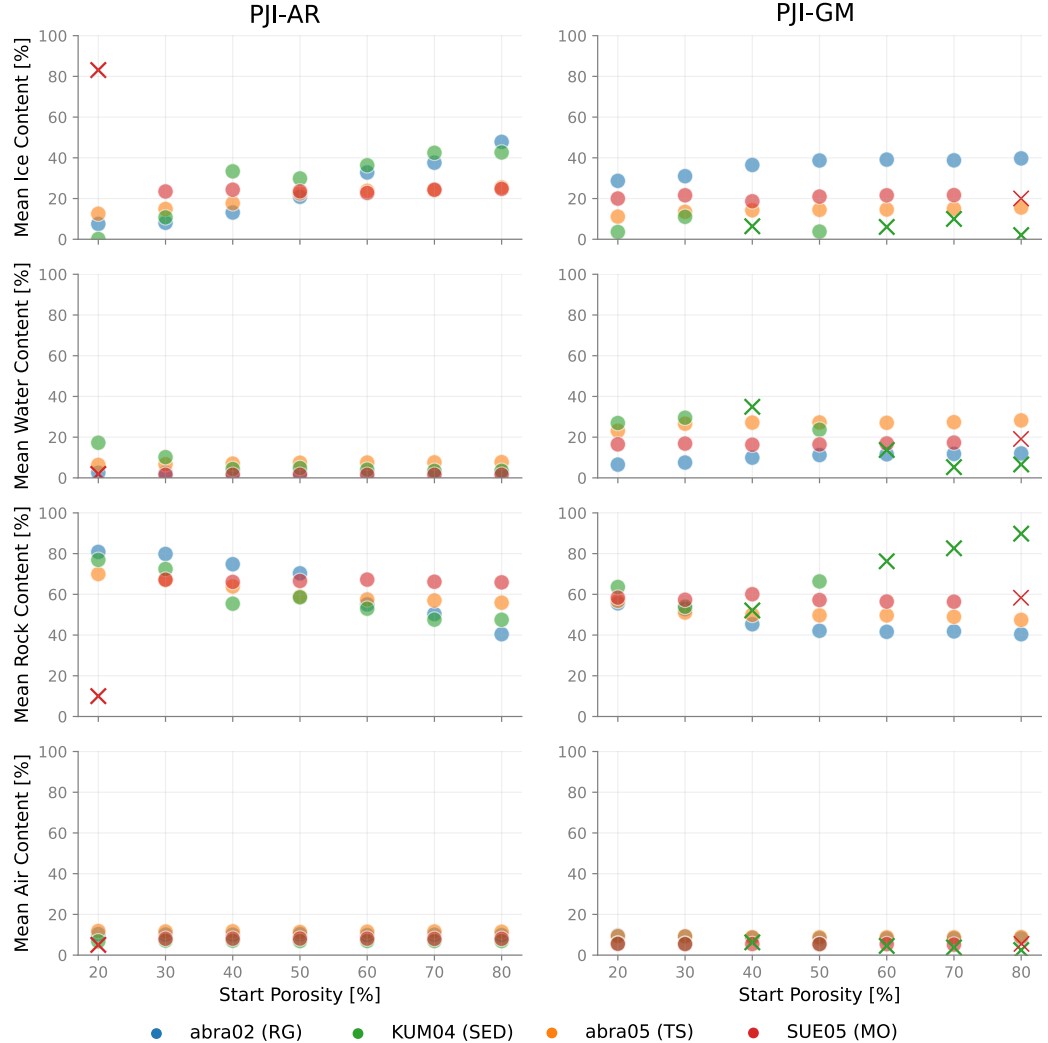

**Figure 15.** Influence of start porosity ($\phi_{start}$) on the modeled volume fractions of ice, water, rock, and air for four profiles that we consider representative for the four different landforms (abra02, abra05, SUE05, and KUM04) for both PJI-AR and PJI-GM model versions. Model runs with a $\chi^2 > 10$, indicating a poor model fit, are marked with crosses in the colors corresponding to the profile and are excluded from further analysis.

(abra02), making PJI-AR more sensitive to the chosen start porosity. In contrast to the PJI-GM model, on the other hand, where minor parameter adjustments can induce substantial $\chi^2$ errors, the PJI-AR model demonstrates greater robustness. Specifically, with the exception of the lowest initial porosity (20 %) for profile SUE05, all other porosities lead to model convergence and acceptable $\chi^2$ values.





**Table 6.** Resistivity values for ice, rock, water, and air used for the porosity sensitivity analysis in Figure 15 for all four representative profiles.

| Profile | $\rho_{ice}$ [Ω m] | $\rho_{rock}$ [Ω m] | $\rho_{water}$ [Ω m] | $\rho_{air}$ [Ω m] |
|---------|--------|--------|--------|--------|
| abra02 | 800'000 | 7'000 | 2 | 1'000'000 |
| abra05 | 500'000 | 20'000 | 10 | 1'000'000 |
| SUE05 | 800'000 | 50'000 | 2 | 1'000'000 |
| KUM04 | 50'000 | 2'000 | 2 | 100'000 |

## 5 Discussion

### 5.1 Permafrost ground ice contents in the Tien Shan and Pamir of Central Asia

In this study, we modeled permafrost ground ice contents for different landforms in the Tien Shan and Pamir of Kyrgyzstan and Tajikistan using two versions of the PJI model which employ two different petrophysical relations for electrical resistivity: Archie's Law (PJI-AR) and the Geometric Mean model (PJI-GM). The modeled ground ice contents provide valuable insights into typical ground ice contents in various landforms within the mountain permafrost of Central Asia, where such information is currently lacking. Our results do not indicate significant differences in ground ice contents between the different geographic regions (e.g., Tien Shan versus Pamir Alay or Eastern Pamir), suggesting that ground ice contents are more landform-specific. The mean ground ice content estimates for the different investigated landforms can be summarized as follows, based on our best-guess estimates from both PJI-GM and PJI-AR applied to 6 rock glaciers, 3 moraines, 4 talus slopes and 9 sediment profiles: Rock glaciers showed a ground ice content of 38–60 %, moraines ranged from 18–40 %, talus slopes from 20-40 %, while fine-grained sediments contained 0–20 %.

The results suggest the presence of ground ice in all investigated landforms, including most of the fine-grained (vegetated) sediment profiles, as confirmed by the borehole core (shown in Fig. 16). The profile at the borehole (KUM04) is situated on a high-altitude plateau with a mean altitude of around 3540 meters above sea level. Surface conditions in this region resemble those of the Tibetan Plateau (e.g. Buckel et al., 2020; Gao et al., 2016; You et al., 2017), where numerous thermokarst lakes indicate the potential for widespread ice-saturated conditions in various states of degradation, especially pronounced in area 2 marked in Fig. 16. While our modeling and observations at the borehole show saturated conditions with relatively low ice contents (20 %), it is possible that ice-rich conditions are more prevalent near these thermokarst lake areas. However, due to the lack of RST measurements and, consequently, PJI results in this specific area, further confirmation is required. Fig. ?? provides an overview of this region, located south of Lake Issyk-Kul in the Central Tien Shan of Kyrgyzstan. If we were to upscale the estimated ground ice contents at KUM04 and consider uniform subsurface conditions within section 1 of Fig. ??, which encompasses approximately 12 square kilometers, with a 5-meter-deep layer of permafrost containing 20 % ice content, this specific area alone would contain an estimated volume of 120,000 m³ of ground ice. The presence of such ice-saturated conditions in fine-grained sediments contrasts with similar environments in the European Alps, where such sediments



are generally believed to lack permafrost (Hoelzle, 1994; Kenner et al., 2019). These ice-rich conditions, despite the absence of surface geomorphological indicators, underscore the importance of geophysical methods in detecting and characterizing permafrost in regions where visual identification is challenging. This is particularly critical for high-altitude plateaus like those in Central Asia, which host significant infrastructure, such as the Kumtor gold mine, and where ongoing permafrost degradation

could lead to considerable ground instability and infrastructure risks, as evidenced by the formation of thermokarst lakes in the region.

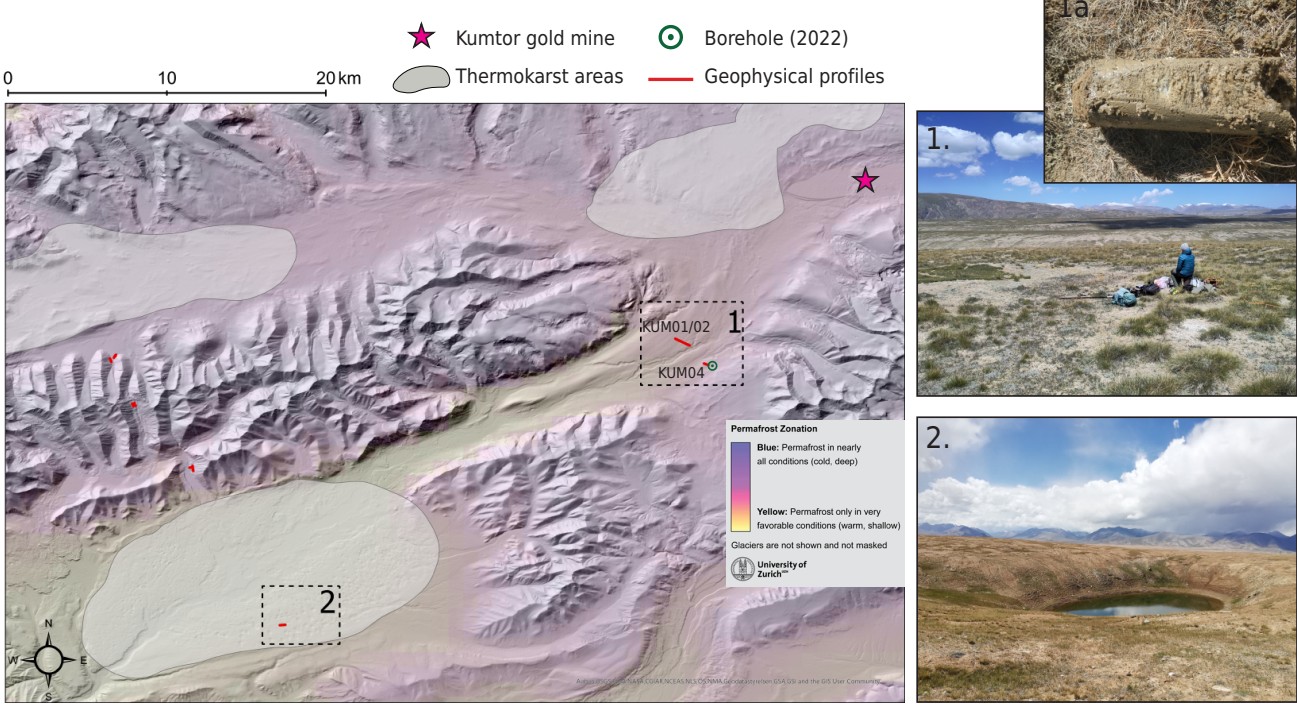

**Figure 16.** Map showing one of the regions where ground ice is probably extensive (Gruber, 2012) in a high-altitude plateau with fine-grained, partly vegetated sediments. Sources of the background hillshade map: Esri, DigitalGlobe, GeoEye, i-cubed, USDA FSA, USGS, AEX, Getmapping, Aerogrid, IGN, IGP, swisstopo, and the GIS User Community

    The mean ground ice contents quantified for rock glaciers (38 - 60 % ) in our study region fall within the range reported by other studies, where ground ice contents were either empirically estimated or quantified using the 4PM or PJI approaches in the Andes (e.g. Rangecroft et al., 2015; Jones et al., 2019; Schaffer et al., 2019; Jones et al., 2021a; Mathys et al., 2022;

Halla et al., 2021; Hilbich et al., 2022), or in the European Alps (Mollaret et al., 2020; Duvillard et al., 2018; Pavoni et al., 2023). Furthermore, drill cores taken from rock glaciers confirm the potential for high ice contents within these landforms (e.g. Vonder Muhll and Haeberli, 1990; Vonder Mühll and Holub, 1992; Monnier and Kinnard, 2013; Krainer et al., 2015). For permafrost landforms other than rock glaciers, data are more scarce. In this study, we found mean ground ice content



in fine-grained sediment profiles of approximately 20 %, similar to those found in the Andes by Hilbich et al. (2022), using the 4PM (Hauck et al., 2011). Quantitative data on ice content in talus slopes is similarly limited (Scapozza et al., 2015). However, the results from a Swiss Alps talus slope study by Mollaret et al. (2020), indicating ground ice contents of 30-40 %, align with our findings of 20-40 %. Although many studies have addressed the presence of ice-rich moraines (e.g. Bolch et al., 2019), quantitative ground ice content estimates remain largely absent. A study conducted by Kunz et al. (2022) using the 4PM recently estimated ice contents to be around 40 % in a moraine located in the European Alps, which is within the range of our estimates of 18-40 %. However, our study suggest a significant variability that extends previous estimates for this landform, not just in between different mountain ranges and landforms but also within individual locations (Fig. **??**). These comparisons provide significant insights into the variability and uniformity of ground ice content across various mountain permafrost landforms in different mountain ranges.

Finally, the presence of ground ice in talus slopes, moraines, and fine-grained sediments at all study sites underscores the need to consider permafrost landforms beyond rock glaciers. This is especially important when evaluating ground ice content and associated water equivalents to gauge the hydrological significance of permafrost (degradation) in different regions (e.g. Azócar and Brenning, 2010; Jones et al., 2019, 2018). In mountain hydrology, the role and importance of permafrost is especially poorly understood and due to the lack of supporting data this leads to an assumed negligible importance for the hydrological cycle (van Tiel et al., 2024). The estimates presented here are therefore an important step to fill a knowledge gap about the cryosphere groundwater connectivity. In Central Asia, obtaining more validation data would significantly improve estimates of ground ice contents for the region. However, conducting extensive borehole drilling for this purpose is impractical due to logistical challenges and financial implications. This renders non-invasive geophysical methods, despite the uncertainties outlined in this study, even more importance.

## 5.2 Evaluation of the PJI-GM for different landforms

The ground ice content estimates presented were modeled using two different versions of the PJI approach. We evaluated the suitability of the PJI-Geometric Mean model, which has not been extensively tested previously, for quantifying ground ice contents across various landforms. While the Geometric Mean model can generally be applied to all the sampled landforms using the proposed methodology, differences were observed between the landforms and across multiple model runs in terms of the estimated ground ice contents and their spatial distribution within each profile. This is exemplified by the distinct PJI-GM clusters identified, particularly for more fine-grained landforms.

The PJI-GM model appears to be most suitable for profiles with a distinct ice-rich layer, as observed in rock glaciers and ice-rich moraines. The examination of ERT tomograms indicated that all rock glaciers display a similar resistivity distribution pattern, characterized by an active layer overlying a thick, high-resistivity layer, which differentiates them from most other profiles where the spatial resistivity distribution is more variable across the longitudinal profile. In these cases, the ambiguity between different PJI-GM cluster results and derived subsurface ground ice contents is minimal. This specifically good performance of the PJI-GM for ice-rich permafrost profiles compared to the PJI-AR model is likely attributable to the fact that





the PJI-AR model does not include the resistivity of rock and ice, so that the ice-rock ambiguity is solely constrained by the seismic (RST) data.

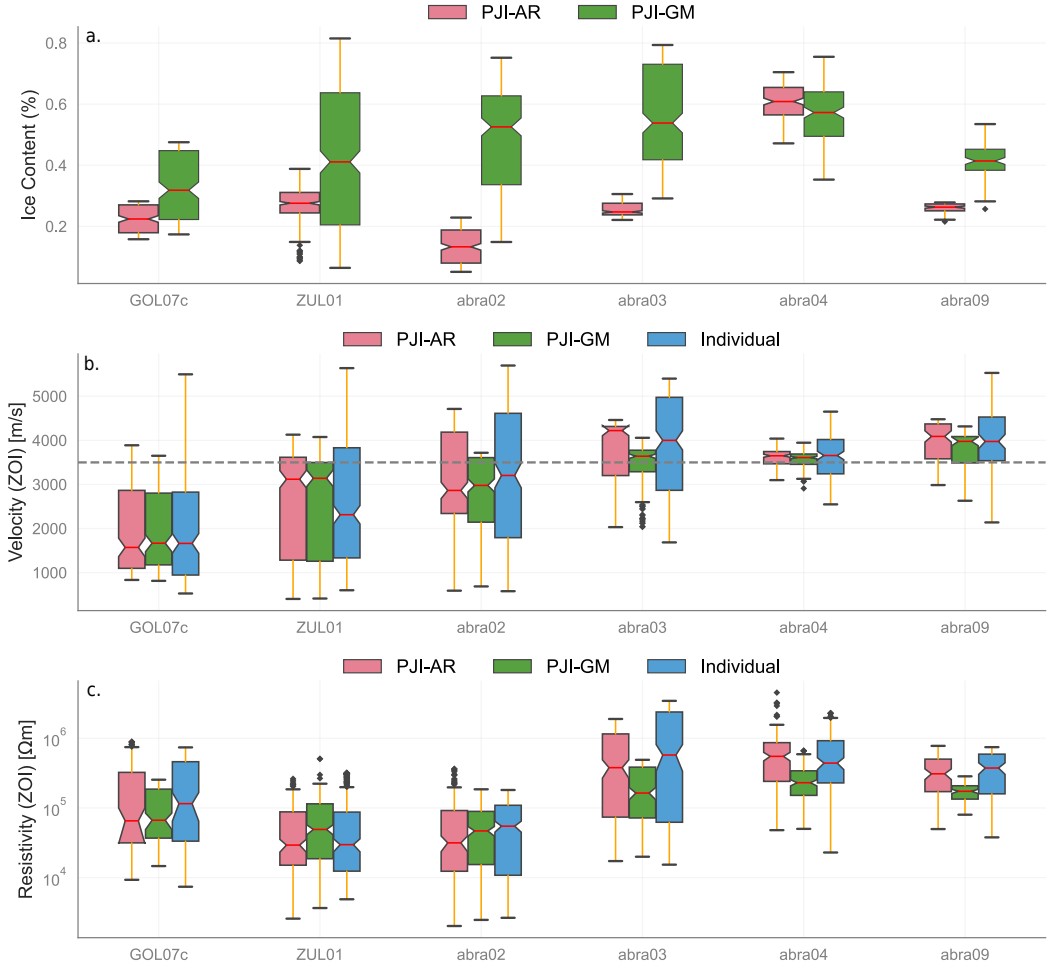

**Figure 17.** (a) Ice content boxplots for all rock glaciers with PJI-GM and PJI-AR, extracted from a representative zone (ZOI); (b) P-wave velocity boxplots of each profile from the same ZOI (PJI-AR, PJI-GM and individual inversion); (c) Resistivity boxplots of each profile from the same ZOI (PJI-AR, PJI-GM and individual inversion). The dashed line in (b) marks the P-wave velocity of ice (3500 m/s)

The PJI-AR version tends to underestimate ice contents of presumably ice-rich landforms. As discussed before, the only
exception, where PJI-AR produces similar results for a rock glacier site as the PJI-GM is profile abra04. In this specific profile, a massive ice-rich layer was observed through outcrops in the field. Figure 17b shows that the P-wave velocities (both measured and modeled) for this profile are close to the P-wave velocity of ice (3500 $m/s$, marked by the dashed line in Figure 17) with little spatial variability, which sets this profile apart from other rock glacier profiles where mean P-wave velocities are usually either higher or lower, or show a larger spread within the representative zone (ZOI). These results suggest that rock glacier




abra04 may contain a core of pure ice from remnant glacier ice, distinguishing it from the other sampled rock glaciers. For this profile, combining Archie's law with Timur's equation likely helps to model the higher ice content, even though electrolytic conduction is improbable. This contrasts with other rock glaciers where P-wave velocities are typically higher; closer to those of rock than ice. In this case, the PJI-AR has more difficulties to distinguish between rock and ice. Interestingly, even when seismic velocities exhibit only minor fluctuations around the $3500 \, \mathrm{m \, s^{-1}}$ threshold, as observed in profile abra09 or abra03,

PJI-AR results converge to corresponding ground ice contents significantly lower than PJI-GM. Therefore, we would prioritize using the PJI-GM model for rock glacier profiles, where heterogeneous compositions are likely, especially if no boreholes or other information about the subsurface is available.

For less ice-rich landforms with a lack of a distinct high-resistivity layer and where profiles exhibit more variable and generally lower measured resistivities, the PJI-GM produces ambiguous results, with clusters that strongly differ regarding the

modeled subsurface conditions. The resulting subsurface ice content models vary hereby significantly across different clusters (cf. Fig. 10). For talus slopes and certain moraines analyzed, the fine to medium-grained materials may facilitate increased subsurface water storage. This higher water content within these landforms may lead to enhanced electrolytic conduction, possibly accounting for the more comparable results between the PJI-GM and PJI-AR models. While the modeled talus slopes and some moraines in this study exhibited relatively fine-grained materials, this may not be representative of the general grain

size characteristics of these landforms across different sites. Talus slopes and moraines can display significant variability in grain size, often featuring coarse-blocky compositions comparable to rock glaciers, depending on factors like local geology. In such cases, where coarser materials dominate, the PJI-AR model might also underestimate ice content, similar to the limitations observed for rock glaciers. This suggests that the PJI-GM model may be more appropriate for these coarser-grained settings. Unfortunately, the available data from this study does not allow for confirmation of this assumption.

The PJI-GM model exhibited more ambiguous performance when applied to fine-grained sediment profiles. For many SED profiles, it was challenging to find parameter combinations that led to successful inversion convergence. Numerous runs resulted in either a lack of convergence, characterized by high RMS and $\chi^2$ errors, as also described in Mollaret et al. (2020), or produced unrealistic tomograms with improbable subsurface fraction distributions. This difficulty in achieving reliable results with the PJI-GM for landforms with lower resistivity, such as the SED profiles examined in this study, highlights a potential

limitation of the PJI-GM. However, selecting lower rock resistivity values ($\rho_r$), below the minimum threshold of $1000 \, \Omega m$ used in this study, could potentially help reduce the unrealistic water content estimations observed in some of those profiles. Furthermore, estimating ground ice content in fine-grained sediments at our study sites is inherently challenging due to the absence of clear surface expressions like outcrops or gelifluction features, which are typically absent from flat terrain (Matsuoka, 2001). Given these factors, the PJI-AR model might be generally better suited for analyzing fine-grained sediment profiles,

particularly when a priori knowledge of ice/rock content is available. This can also be justified by the higher probability of electrolytic conduction being the dominating electrical conduction process in these fine-grained sediment profiles (e.g Mele et al., 2014). Furthermore, our findings indicate that the PJI-GM has a tendency to overestimate water content in certain talus slope, sediment, and moraine profiles for layers with low resistivities. This leads to unrealistic water content estimates in the tomograms (> 50 - 60 %) in layers with low resistivities, as frequently observed in the uppermost layers of our fine-grained



sediment profiles (active layer). This overestimation of water content by the PJI-GM in low resistivity layers is likely due to the geometric mean calculation heavily weighting the low resistivity of water (see Eq. 4). Each component's resistivity ($\rho_r$, $\rho_i$, $\rho_w$, $\rho_a$) is raised to the power of its respective fraction. Due to the significantly lower resistivity of water ($\rho_w$), the geometric mean calculation tends to overemphasize its contribution in low resistivity layers, leading to an overestimation of water content.

While constraining maximal water fractions within the PJI-GM model could potentially mitigate this overestimation, our
initial attempts revealed a tendency for such constraints to hinder inversion convergence. One potential solution to this issue could be to include a penalty function or modify the weighting scheme in the geometric mean calculation to reduce the disproportionate influence of low resistivity phases like water. This approach would involve adjusting the model to more accurately reflect the physical conditions of the subsurface, potentially improving the reliability of the PJI-GM model in low-resistivity environments. Furthermore, accounting for surface conduction, as noted by Mollaret et al. (2020) and Steiner et al.
(2021a), for example, might further improve the results.

### 5.3    Uncertainties in the ground ice content quantification of this study

The uncertainty of the quantification of the ice content presented depends on several factors, which are discussed below:

(i)   Standard uncertainties of the geophysical data such as the general ERT and RST data quality, resolution capacity and investigation depth of the surveys, as well as potential inversion artefacts can all impact the individual inversions, as
well as the PJI modeling results. Poor data quality leads to errors in the inversion process, impacting the accuracy of the resulting subsurface models (Hilbich et al., 2009; Hauck et al., 2011; Hilbich et al., 2022; Wagner et al., 2019; Mollaret et al., 2020).

(ii)   The PJI model relies on numerous parameters that can influence the model output and, consequently, the estimated ground ice content. For example, Mollaret et al. (2020) and Wee et al. (2024) (pre-print) suggest that higher start porosi-
ties typically lead to higher ice contents. However, our findings indicate that the initial porosity does not systematically impact the resulting ground ice content, particularly when using the PJI-GM model. Here, both increasing, decreasing, and indifferent ice content were observed with increasing start porosity, with maximum changes in ice content of 10 % (Fig.15). In our study, we found that the start porosity has a larger impact on PJI-AR results than on PJI-GM results (maximum changes of 45 % for the rock glacier profile abra02). Furthermore, incorporating constraints on subsurface
fractions, such as limiting the range of rock content (porosity) or water content, could further enhance the model accuracy. This can be particularly useful when detailed a priori information about the subsurface is available. Constraining the rock content can be particularly valuable in cases where the unconstrained model struggles to accurately reproduce observations or when significant ambiguity exists between PJI-GM clusters. For example, PJI-GM overestimates ground ice content in profiles like SUE02 (Cluster 2) and #457 (Cluster 3), leading to unrealistic tomograms for the rock fraction.
While adding constraints on subsurface fractions could improve model accuracy, this requires detailed a priori knowledge of subsurface conditions, which is often unavailable for remote study sites lacking borehole data. Due to the remote



locations studied and the intention to assess the model without bias, we opted to not systematically apply additional constraints on the subsurface fractions.

(iii) Equation 2, which links P-wave velocities to the volumetric fractions of ice, rock, air, and water, may not be sufficiently accurate for our study. Our findings indicate that very high P-wave velocities (> 5000 m/s) are not well reproduced in the transformed RST tomograms across most profiles and landforms. Incorporating a more sophisticated petrophysical relationship for the seismic data could further enhance the accuracy of estimating the four subsurface phases, leading to more reliable ground ice content quantification.

(iv) The lack of ground truth data presents a significant challenge in validating ground ice content results. Our hierarchical clustering approach for analyzing the PJI-GM model outputs mitigates some uncertainties by showing different possible subsurface ice content distributions, which varied in ambiguity depending on the landform. This method allowed for the exclusion of clusters with unrealistic tomograms in any of the subsurface fractions or transformed ERT or RST tomograms. Additionally, the correlation analysis used to select the final best-guess cluster in most cases coincided with what we would have estimated based on expert knowledge alone. This approach effectively aligns clusters with expert knowledge for most profiles and eliminates implausible results, although it still depends on initial assumptions and the quality of the input data. The variability observed among the plausible clusters (which were not excluded because of unrealistic tomograms in any of the subsurface fractions) can be leveraged to establish uncertainty ranges for the estimated ground ice contents.

## 6 Conclusions

This study provides a comprehensive electrical and seismic dataset that offers significant insights into permafrost occurrences and ground ice contents in various landforms within the Tien Shan and Pamir regions of Central Asia. By evaluating the applicability of the PJI-GM model for quantifying ground ice contents from geophysical data, we highlighted its strengths and limitations across different landforms.

Our findings underscore the effectiveness of the PJI-GM model for modeling distinct subsurface structures, particularly in ice-rich landforms such as rock glaciers. The PJI-GM model outperforms the formerly used PJI-AR model in these contexts, as it more accurately reflects high ice contents, and sharp low-to-high resistivity transitions with minimal ambiguity between model runs. This indicates that the PJI-GM model is particularly suited for characterizing ice-rich landforms with layers of high resistivity, making it a more reliable tool for these specific applications. However, the PJI-GM encounters challenges with fine-grained sediments, where reduced model convergence and overestimation of water content were observed. In these cases, the PJI-AR model might be more appropriate, especially when extensive parameter tuning is not feasible.

The mean estimates of ground ice content for the various sampled landforms in our study sites in Central Asia can be summarized as follows: rock glaciers exhibit ice contents ranging from 38-60 %, moraines from 18-40 %, talus slopes from



20-40 %, and fine-grained sediments from 0-20 %. These results emphasize the necessity of comprehensive assessments across different permafrost landforms to achieve accurate ground ice estimations, beyond just focusing on rock glaciers.

The impact of this study extends beyond the immediate findings. The baseline dataset provided can be instrumental in future monitoring of permafrost dynamics in Central Asia, especially in the context of climate change. Furthermore, this dataset is valuable for modeling studies that aim to assess the hydrological impacts of permafrost thaw and for conducting hazard assessments or adaptation planning for infrastructure built on permafrost.

In conclusion, the PJI-GM model proves to be a valuable tool for specific landforms, particularly ice-rich environments, but

requires careful parameter selection for other landforms to ensure accurate results. The insights and baseline data from this study contribute significantly to the understanding of permafrost in Central Asia, laying the groundwork for future research, monitoring efforts, and climate impact assessments. These efforts aim to initiate and partially sustain future endeavors to establish a long-term monitoring network for the Essential Climate Variable (ECV) permafrost, crucial for understanding and mitigating the impacts of permafrost degradation on infrastructure and hydrology in the region.



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

(PAMIR)' (grant number: SPI-FLAG-2021-001) funded by the SPI Flagship Initiative of the Swiss Polar Institute and the project 'Strengthening the resilience of Central Asian countries by enabling regional cooperation to assess glacio-nival systems to develop integrated methods for sustainable development and adaptation to climate change' funded by the Global environment Facility / United Nations Development Programme / United Nations Educational, Scientific and cultural Organization (GEF/UNDP/UNESCO, contract no. 4500484501). The improvement of the PJI was supported by the Swiss National Science Foundation (grant no. 200021L_178823). We extend our sincere gratitude

to all the field helpers for their invaluable assistance. We also express our deep appreciation to our local partners, including the Central Asian Institute for Applied Geosciences (CAIAG), Kyrgyz Hydrometeorological Service (Kyrgyz Hydromet), the Centre for Research of Glaciers of the National Academy of Sciences of Tajikistan (CRG), and the Aga Khan Agency for Habitat (AKAH). Their support was crucial for the successful collection of data, and this research would not have been possible without their contributions.



## Appendix A: Appendix



**Figure A1.** ERT tomograms (Wenner array) of all profiles, sorted by the study sites. Profile information and statistics are summarized in Table A1.





**Figure A2.** Individual ERT and RST inversion results and best-guess PJI-GM ice content estimates for all profiles. The zone of interest (ZOI), over which mean values were calculated, is drawn in white dotted rectangles. If a profile spans more than one landform, there are multiple ZOIs, one per landform.



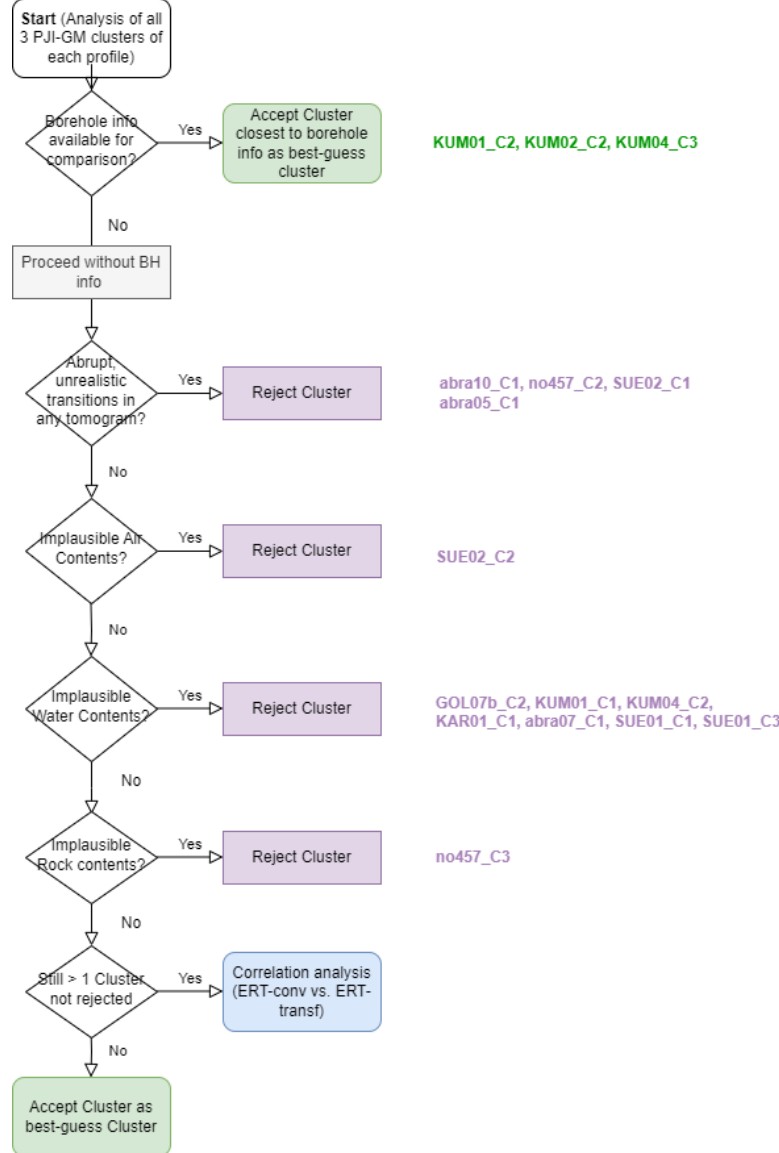

**Figure A3.** Flowchart depicting the analysis process for selecting the best-guess cluster among the PJI-GM clusters of each profile. The process begins with checking the availability of borehole (BH) information. If BH information is available, the cluster closest to the borehole information is accepted as the best-guess cluster (highlighted in green: KUM01_C2, KUM02_C2, KUM04_C3). If BH information is not available, the analysis proceeds without it, rejecting clusters based on various criteria: abrupt unrealistic transitions (rejected clusters: abra10_C1, no457_C2, SUE02_C1, abra05_C1), implausible air contents (rejected cluster: SUE02_C2), implausible water contents (rejected clusters: GOL07b_C2, KUM01_C1, KUM04_C2, KAR01_C1, abra07_C1, SUE01_C1, SUE01_C3), and implausible rock contents (rejected cluster: no457_C3). If multiple clusters remain after these rejections, a correlation analysis between ERT-conv and ERT-transf is conducted to determine the best-guess cluster.



**Figure A4.** Boxplots showing the distribution of geophysical properties for the zone of interest (ZOI) in all profiles. The first subplot illustrates the ice content extracted from the ZOI. The second subplot presents the P-wave velocities corresponding to the same ZOI. The third subplot shows the resistivity values within the ZOI. Each subplot highlights the variability and range of values for the respective property for all profiles.





**Table A1.** Summary of ERT (Electrical Resistivity Tomography) filtering details for various profiles, including the profile year, landform type, array spacing, profile length, the number of filtered points, the total number of data points, the percentage of data points remaining after filtering, and the Root Mean Square Error (RMSE) percentage. For the electrode array: W = Wenner, DD = Dipole-Dipole.

| Profile | year | landform | array | spacing | profile length (m) | # filtered points | # data points total | % data points remaining after filter | RMSE (%) |
|---|---|---|---|---|---|---|---|---|---|
| abra01 | 2021 | MO | W | 5 | 235 | 5 | 360 | 98.61 | 1.67 |
| abra02 | 2021 / 2022 | RG / SED | W | 5 | 595 | 105 | 1188 | 91.16 | 3.77 |
| abra03 | 2021 / 2022 | RG / SED | W | 5 | 475 | 27 | 912 | 97.04 | 8.58 |
| abra04 | 2021 | RG | W | 5 | 355 | 4 | 636 | 99.37 | 2.75 |
| abra05 | 2021 / 2022 | TS | W | 5 | 235 | 3 | 360 | 99.17 | 5.18 |
| abra06 | 2021 | TS / RG | W | 5 | 235 | 7 | 360 | 98.06 | 3.82 |
| abra07 | 2021 | SED | W | 5 | 235 | 0 | 360 | 100.0 | 3.72 |
| abra08 | 2021 | MO | W | 5 | 235 | 25 | 360 | 93.06 | 2.23 |
| abra09 | 2022 | RG | W | 5 | 355 | 72 | 636 | 88.68 | 3.53 |
| abra10 | 2022 | MO | W | 5 | 235 | 6 | 360 | 98.33 | 2.62 |
| BUL01 | 2023 | SED | W | 2 | 94 | 0 | 360 | 100.0 | 1.82 |
| GOL01 | 2021 / 2022 | TS | W | 4 | 235 | 2 | 360 | 99.44 | 1.85 |
| GOL01 | 2022 | TS | DD | 5 | 235 | 217 | 944 | 77.01 | 28.99 |
| GOL02_H | 2021 | RG | W | 5 | 235 | 25 | 360 | 93.06 | 11.23 |
| GOL02_V | 2021 / 2022 | RG | W | 5 | 235 | 10 | 360 | 97.22 | 4.76 |
| GOL05 | 2022 | MO | W | 5 | 235 | 6 | 360 | 98.33 | 5.58 |
| GOL05 | 2022 | MO | DD | 5 | 235 | 48 | 944 | 94.92 | 7.25 |
| GOL06 | 2022 | RG | W | 5 | 235 | 7 | 360 | 98.06 | 2.75 |
| GOL07 | 2022 | SED / MO / RG | W | 5 | 595 | 53 | 1188 | 95.54 | 3.2 |
| GOL08 | 2022 | TS | W | 5 | 235 | 7 | 360 | 98.06 | 5.71 |
| KUM01 | 2022 | SED | W | 5 | 235 | 12 | 360 | 96.67 | 3.33 |
| KUM01 | 2022 | SED | DD | 5 | 235 | 8 | 944 | 99.15 | 9.6 |
| KUM02 | 2022 | SED | DD | 5 | 835 | 545 | 5664 | 90.37 | 5.31 |
| KUM04 | 2022/2023 | SED | DD | 5 | 235 | 11 | 944 | 98.83 | 10.25 |
| KUM04 | 2022/2023 | SED | DD | 2 | 94 | 2 | 944 | 99.79 | 11.11 |
| no599 | 2021 | MO | W | 3 | 285 | 21 | 912 | 97.7 | 12.87 |
| no457_01 | 2023 | SED | W | 5 | 235 | 8 | 360 | 97.78 | 2.26 |
| no457_01 | 2023 | SED | DD | 5 | 235 | 108 | 1188 | 90.91 | 2.66 |
| no457_02 | 2023 | RG | W | 5 | 235 | 59 | 360 | 83.61 | 7.65 |
| SUE01 | 2021 / 2023 | TS | W | 3 | 210 | 60 | 636 | 90.57 | 3.83 |
| SUE01 | 2021 / 2023 | TS | DD | 3 | 210 | 258 | 944 | 72.67 | 33.3 |
| SUE02 | 2022 | TS | DD | 5 | 235 | 86 | 944 | 90.89 | 4.54 |
| SUE02 | 2022 / 2022 | TS | W | 5 | 235 | 10 | 360 | 97.22 | 3.12 |
| SUE03 | 2021 | RG / TS | W | 5 | 355 | 8 | 360 | 97.78 | 2.76 |
| SUE03_V | 2021 | TS / RG | W | 5 | 235 | 56 | 636 | 91.19 | 3.39 |
| SUE04 | 2021 | SED | W | 5 | 235 | 8 | 360 | 97.78 | 3.41 |
| SUE05 | 2023 | MO | W | 5 | 235 | 4 | 360 | 98.89 | 2.4 |
| SUE05 | 2023 | MO | DD | 5 | 235 | 14 | 944 | 98.52 | 7.02 |
| SUE06 | 2023 | MO | DD | 5 | 235 | 44 | 944 | 95.34 | 3.44 |
| yak01 | 2022 | RG | W | 5 | 235 | 23 | 360 | 93.61 | 4.54 |
| yak02 | 2022 | RG | W | 5 | 235 | 35 | 360 | 90.28 | 6.64 |
| ZUL01 | 2023 | SED / RG | W | 5 | 355 | 44 | 636 | 93.081 | 8.44 |
| ZUL02 | 2023 | MO | W | 5 | 235 | 7 | 360 | 98.05 | 3.03 |
| ZUL02 | 2023 | MO | DD | 5 | 235 | 138 | 944 | 85.38 | 17.7 |
| ZUL03 | 2023 | SED | DD | 5 | 235 | 148 | 1035 | 85.70 | 7.03 |
| ZUL03 | 2023 | SED | W | 5 | 235 | 3 | 360 | 99.16 | 3.8 |
| KAR01 | 2023 | SED | W | 5 | 235 | 4 | 360 | 98.89 | 7.88 |
| KAR01 | 2023 | SED | DD | 5 | 235 | 57 | 944 | 93.96 | 5.98 |



**Table B1.** Summary of RST filtering metadata, including the profile, year, spacing, length, RMSE, $\chi^2$, and percentage of reliable first arrival picks.

| Profile | Year | Spacing (m) | Length (m) | RMSE (m/s) | $\chi^2$ (%) | % of reliable picks |
|---------|------|-------------|------------|------------|--------------|---------------------|
| abra02 | 2022 | 5 | 205 | 0.94 | 0.89 | 94 |
| abra03 | 2022 | 5 | 205 | 1.12 | 0.81 | 93 |
| abra04 | 2022 | 5 | 205 | 1.48 | 0.98 | 96 |
| abra05 | 2022 | 5 | 115 | 1.28 | 0.86 | 90 |
| abra07 | 2022 | 5 | 115 | 1.15 | 0.91 | 96 |
| abra09 | 2022 | 5 | 115 | 0.88 | 0.77 | 88 |
| abra10 | 2022 | 5 | 115 | 1.40 | 0.87 | 89 |
| BUL01 | 2023 | 2.5 | 46 | 1.47 | 0.97 | 92 |
| GOL01 | 2022 | 2.5 and 5 | 115 | 1.25 | 0.79 | 85 |
| GOL07a | 2022 | 5 | 115 | 1.70 | 1.29 | 89 |
| GOL07b | 2022 | 5 | 115 | 1.36 | 0.82 | 90 |
| GOL07c | 2022 | 5 | 115 | 1.42 | 0.90 | 94 |
| KAR01 | 2023 | 5 | 115 | 1.15 | 0.79 | 91 |
| KUM01 | 2022 | 2 and 5 | 115 | 1.21 | 0.88 | 90 |
| KUM02 | 2022 | 2 and 5 | 115 | 1.49 | 0.99 | 90 |
| KUM04 | 2022 | 2.5 and 5 | 115 | 1.38 | 0.85 | 95 |
| SUE01 | 2023 | 3 | 69 | 1.31 | 0.76 | 83 |
| SUE02 | 2023 | 5 | 115 | 0.61 | 1.41 | 88 |
| SUE05 | 2023 | 5 | 115 | 0.81 | 1.02 | 93 |
| no457 | 2023 | 5 | 115 | 0.65 | 1.69 | 89 |
| ZUL01 | 2023 | 5 | 115 | 0.59 | 1.41 | 86 |
| ZUL02 | 2023 | 5 | 115 | - | - | 71 |