# Peer review of "Quantifying permafrost ground ice contents in the Tien Shan and Pamir (Central Asia): A Petrophysical Joint Inversion approach using the Geometric Mean model"

_EGUsphere, 2024_

## Author Comment (AC1)

**Reviewer 2 comments and author answers:**

The paper presents an extensive dataset of geophysical measurements (ERT and RST) collected from various sites across the Tien Shan and Pamir regions. For the interpretation of these datasets, the authors thoroughly explore the potential of petrophysical joint inversion (PJI) using the resistivity geometric mean model (PJI-GM) and partially compare it with the more commonly used PJI based on Archie's Law (PJI-AR).

The study is well-written and provides a detailed and rigorous description of the methodologies employed. However, it lacks a brief discussion comparing the inversion schemes in terms of convergence metrics (e.g., $chi^2$ and/or RMSE). Beyond this observation, I have included additional comments in the manuscript attached to this review and recommend minor revisions or technical corrections to the paper.

Thank you for your positive feedback and for recognizing the rigor of our methodology. Regarding your suggestion to include a discussion comparing the inversion schemes in terms of convergence metrics (e.g., $chi^2$ and RMSE), we have already compiled these error statistics for the four representative profiles shown in Figure 15, as presented in Table 1 below.

A direct comparison is possible for these specific profiles because we report the resistivity values of all phases for single runs in Table 6, allowing for a 1:1 comparison between PJI-AR and PJI-GM. However, for the remaining profiles, such a comparison is not straightforward. The PJI-GM results are based on multiple inversion loops with varying resistivity values, from which a mean was used to define each cluster. In contrast, for PJI-AR, we did not systematically loop over different parameters. As a result, comparing a mean RMSE and $chi^2$ for PJI-GM with single-run results from PJI-AR would not be meaningful.

That being said, for the four representative profiles, the error metrics do not show significant differences between PJI-AR and PJI-GM. Given this, we do not believe an extended discussion of convergence metrics is absolutely necessary if the reviewer does not insist.

LN 131: It is a bit confusing that you cite profiles ERT that you won't discuss in the text. Either clarify here that this study only focused on the profiles where you got both ERT and RST measurements or simply quote the measurements you'll presents further on.

Thank you for your observation regarding the use of ERT profiles in the study. While the primary focus of the joint inversion analysis is on profiles where both ERT and RST measurements are available, all ERT profiles are used in the general characterization of permafrost conditions within the study region (e.g. Fig. 7). We believe including all ERT profiles is important to provide a comprehensive view of the dataset, highlighting the significant amount of new information gathered in a region where data availability was previously limited. Additionally, all ERT profiles are provided in the Annex (Fig. A1) to ensure transparency and accessibility for future studies. Therefore, we prefer to leave the mention of all data here as is. The same goes for the comment about Figure 1, where the reviewer suggests to only show the sites where both ERT and RST are available. Most sites (except for Yakarcha and no 599) have both ERT and RST data, However, we also use the data from those two sites to characterize the general permafrost conditions in the region, as mentioned above (chapter "Permafrost characteristics of the different sites and landforms in Central Asia, LN375-414 of the original manuscript).

Fig 1: In this small box-map I suggest to increase the label font in order to read the name of the different countries represented

We completely agree and increased the size of the box-map fonts for better readability.

Caption of Table 1: doesn't red too clearly maybe better to reformulate

We changed the caption part that was marked as not clear to:

MAAT (Mean Annual Air Temperature) and mean annual precipitation values were derived from meteorological station (MS) data located near the geophysical measurement sites. These stations are described in Hoelzle et al. (2017) and Schoene et al. (2013).

Table 2: I suggest to add one more column with complementary information available (e.g., previously drilled bore-holes as cited in the text)

Thank you for your suggestion to add a column with complementary information, such as previously drilled boreholes. As there is only one borehole available in the dataset, and it is already mentioned in the column "PF Observations," we believe adding another column would not provide much additional value. We added the following sentence to the table caption and we hope this is OK with the reviewer, but we remain open to further discussion if needed:

*Only the Kumtor site has previous borehole data that was available for comparison for this study.*

Figure 3: Would be best to either increase text in the figure or maybe change color (I'm referring to the line-labels)? cause at the moment it is hard to read. Also I think either here or in Figure 1 you should add the location of the MS you quote the data from within the text.

This is an excellent suggestion, and we appreciate your input to enhance the clarity and usability of the figures. We will increase the text size of the line labels to improve readability and add the locations of the meteorological stations (MS) in the figure. Some MS are more than 10 km away from the profile locations. When zooming out so far, the profile lines were not visible anymore in a meaningful way. This is why, for those cases, we marked the direction and distance from the profiles of the closest MS with an annotated arrow in the figure. We hope this is satisfactory.

Caption of Table 3: I would change the sentence and state clarity the conditions on resistivity not the physically impossible ones (e.g. vondition on rho: rho_a>rho_i>rho_r)

We changed the figure caption according to the comment:

Values for rho_i, rho_r, and rho_w tested in the PJI-GM loop. Combinations were limited to those meeting the condition rho_a > rho_i > rho_r for physical plausibility. Units are in Ohm meter (Ohm m).

Table 2, comment about values of rho_w: See other comment in the result part but I would like some references/explanation for the values of water resistivity below 100 Ohm m (they reflects quite the mineralization especially considering 2 Ohm m)

Thank you for pointing out the need for an explanation regarding the values of pore water resistivity below 100 Ωm. While strong mineralization is plausible in the region due to its geological context, we currently lack specific references to confirm this for our specific study sites (Lebedeva et al., 2024). However, it is important to note that freezing leads to the accumulation of ions in the remaining unfrozen pore fluid, thereby lowering the resistivity of the pore fluid. Since our approach considers all four fractions separately, we do not assume bulk resistivity values. As a result, the true resistivity of the pore water might be significantly lower than 100 Ωm. Furthermore, in the Alps, water resistivities lower than 20 Ωm have been measured (Scherler et al., 2010).

LN 315: How? Did you run tests with different values? Or did you review it from literature?

The value of β was selected based on prior studies (Mollaret, 2020, Pavoni, 2023), which indicate that it generally ensures satisfactory mass conservation during the inversion process. We did run tests with various values and we observed during these preliminary tests that varying β around the default value of 10'000 did not significantly affect the inversion results for our dataset. Therefore, we opted to use the default value of 10,000, which has been shown in the literature to be effective in similar applications. We acknowledge that a more detailed investigation of β could be valuable but it has been done in Mollaret et al (2020). We added the following sentence to the manuscript:

Initial testing in our study revealed minimal variations in chi$^2$ and RMS values around this default value, attesting low model sensitivity. Similarly, Mollaret et al. (2020) and Pavoni et al. (2023) reported limited variation in chi$^2$ and RMS values for β values near 10'000, supporting its use as a robust default parameter.

LN 324: Are you also considering the limits on the resistivity values expressed in the label of Table 3?

Yes, those conditions were also considered but are not mentioned here, as they were already filtered out during earlier stages. To enhance clarity, we removed part of the parentheses to directly refer the reader to the workflow illustrated in Figure 3. From the figure, it should be more evident which conditions were used for filtering.

Figure 4: Increases labels for a. and b. in the figure and change the x-axis in figure 4a (at the moment the labels are not really readable so they are a bit useless: are those the number of each tomogram or what are those number within the x-axis?)

Also use a different color between the green and the black (change one maybe to red or something with more contrast because in the Dendogram it gets quite similar between them the black and green line)

We completely agree and (i) changed the color from green to red and (ii) increased the labels for a. and b in the figure. The x-axis labels are indeed the number of each tomogram: In this example, there were 53 tomograms available for the clustering after the filtering step. Therefore, the numbers on the x-axis go from 1-53. We left the x-axis as is and just added this information in the caption:

Dendrogram and scatterplot illustrating the hierarchical clustering of PJI-GM model outputs for the abra02 rock glacier profile (here, for the 53 remaining tomograms after the quality check). (a) Dendrogram resulting from the hierarchical clustering of the extracted features. The x-axis represents the number of each tomogram (from 1 to 53) included in the clustering. (b) Scatterplot of mean ice content and the ice content standard deviation with points colored according to their respective clusters, demonstrating the differences in mean ice content and variability among the clusters.

LN 340: How is this distance or dissimilarity compute? L2 norm?

The hierarchical clustering was performed using Ward's method, which minimizes the total within-cluster variance when merging clusters. This method is based on the Euclidean distance (L2 norm) between data points. We have added this clarification to the manuscript:

We performed hierarchical clustering using Ward's method, which minimizes the total within-cluster variance during merging and is based on the Euclidean distance (L2 norm) between data points (e.g. Ogasawara and Kon, 2021).

L 380: Have you tried to do something similar but with the apparent resistivity values of each profile? (I'm curious to know if there is a "resistivity signature" already from the apparent resistivity data)

We did not analyze the apparent resistivity data in a similar manner to identify potential "resistivity signatures." However, we agree that this could be an interesting avenue for future research, as it might reveal additional insights into the characteristics of the profiles even before inversion.

L 494: Isn't it a bit low this water resistivity considering the environment? (shouldn't it be quite "pure" in this kind of environment and therefore more resistive? >=100 Ohm m?) Do you have any reference for this choice? (for instance 2 Ohm m seems pretty mineralized/ salty water to me)

Thank you for your comment regarding the water resistivity values. While we agree that resistivities higher than 100 $\Omega$m could be expected in environments with relatively pure water, the geological context of our study area suggests significant mineralization in certain regions. For instance, visible salt deposits at the surface indicate that groundwater in these areas can be highly mineralized, which would result in lower resistivity values (e.g. Lebedeva et al., 2024; Shokri et al., 2024). For example, TDS (Total Dissolved Solids) data presented by Lebedeva et al., (2024) confirms high mineralization of the water coming from rock glaciers in northern Tien Shan (roughly 73-115 mg/l). Additionally, they also report similar high mineralization values in water samples, taken from rivers and creeks. Furthermore, we refer also to the study of Scherler et al. (2010) from the Alps, where water resistivities as low as 7 $\Omega$m were measured. This supports our use of lower water resistivity values in these areas. We acknowledge that more precise data on local water resistivity/conductivity would improve the accuracy of the model, and we suggest this as an important direction for future fieldwork.

LN 510: Which value is "acceptable"? a chi2 of 10 is quite high for standard geophysical inversion, maybe you could expand a bit about how you got to this threshold

The threshold for acceptable $chi^2$ values in this study was based on guidelines from the literature. Specifically, Mollaret et al. (2020) and Günther et al. (2006) state that $chi^2$ values around 1–5 provide reliable results, avoiding overfitting or underfitting, while Audebert et al. (2014) consider $chi^2$ values up to 10 as still reliable. Additionally, Mollaret et al. (2020) observe that the Geometric Mean model typically results in higher $chi^2$ values compared to other approaches. In our study, the majority of profiles had $chi^2$ values below 5, aligning with these guidelines. We have added a clarification in the manuscript to address this:

A quality check was applied to remove models with large misfits based on the $chi^2$ and RMS thresholds, as outlined in the workflow (Fig. 3). The $chi^2$ threshold was chosen following guidelines from the literature, where values between 1 and 5 are considered reliable for most applications (Günther et al. (2016), and values up to 10 are acceptable in specific cases (Audebert et al., 2014; Mollaret et al., 2020).

Table 6: Same commentary as before: it seems to me these values for the resistivity of water are a bit too low. Also, in this case of such low resistivity for water (2 Ohm m) I think Archie's law is a fair approximation of electrical conduction

Thank you for your comment. We acknowledge that the resistivity values for water ($\rho_w$) in Table 6 may appear low. However, a low $\rho_w$ was necessary to account for the conductive conditions observed in certain profiles, and higher $\rho_w$ values led to unrealistically high water contents. As mentioned before, field observations suggest that salt content can be significant at some study sites, which supports the possibility of lower pore water resistivity values. Another way to achieve similar results would be to lower the rock resistivity ($\rho_r$), but we chose $\rho_w$ as this is directly affected by increased salt contents.

Figure 17: From Figure (b) and (c) it seems like that the PJI-AR inversion reflects values closer to the individual inversion both for velocity and resistivity: I think would be good to comment on it.

Also I miss in this comparison between the two PJI methods how the chi2 and RMSE are changing between the -AR and -GM inversion schemes

Thank you for your interesting comment. The fact that the PJI-AR inversion produces results that are closer to the individual inversions for both velocity and resistivity (for some profiles) compared to the PJI-GM approach can be attributed to the nature of the parameter interdependencies in the two methods. In PJI-AR, resistivity is controlled by Archie's law, which directly links it to only the water and rock fractions. This can result in a stronger constraint that probably keeps the resistivity values more in line with the individual inversion results. In contrast, the PJI-GM approach considers all four fractions—water, ice, air, and rock—simultaneously, which leads to a more complex interplay of constraints. This broader coupling among parameters can cause the PJI-GM results to diverge slightly more from the individual inversions.

For the comparison of $chi^2$ and RMSE between the two methods, we have compiled error statistics ($chi^2$ and RMSE) for the four representative profiles shown in **Figure 15** of the manuscript in Table 1 below. A direct comparison is possible for these four profiles because we mention the resistivity values of all phases for single runs for these four profiles in **Table 6**, allowing for a 1:1 comparison between PJI-AR and PJI-GM. However, for the remaining profiles, this kind of direct comparison is not possible at the moment. For the PJI-GM runs, resistivity values are derived from multiple inversion loops, where different resistivity values were tested, and a mean was used to define each cluster. Since we did not systematically loop over different parameters for PJI-AR, comparing a mean RMSE and $chi^2$ for PJI-GM against single-run results from PJI-AR would not be meaningful. However, for the four representative profiles, it can be seen from the table that the error metrics do not differ much between PJI-AR and PJI-GM.

*Table 1: RMSE and $chi^2$ comparison for the four representative profiles chosen in the manuscript.*

| Profile | RMSE PJI-AR | RMSE PJI-GM | $chi^2$ PJI-AR | $chi^2$ PJI-GM |
|---------|-------------|-------------|----------------|----------------|
| abra02 | 13.68 | 13.15 | 0.93 | 0.95 |
| KUM04 | 6.94 | 6.36 | 0.89 | 1.03 |
| abra05 | 11.56 | 11.87 | 0.89 | 1.32 |
| SUE05 | 13.6 | 15.98 | 2.01 | 2.77 |

L 615: could also use larger water resistivity values has an impact on this?

We acknowledge that larger water resistivity values could potentially impact the results. In our testing, we found that using higher water resistivity values often led to poor convergence of the inversion, which is why we limited our range to a maximum of 150 Ωm. Especially for SED profiles,

higher $\rho_w$ led to a strong and unrealistic overestimation of the water content in the profiles. While we did not systematically test values above this threshold, exploring the impact of larger water resistivity values could be an interesting direction for future studies. It is likely that the observed effects result from a combination of both water resistivity ($\rho_w$) and rock resistivity ($\rho_r$). A dedicated sensitivity study focusing solely on these two parameters would be very useful, as those hold clear hydrogeological significance. However, within our current approach—where we rely on looped inversions and use numerical inversion parameters such as chi$^2$ and RMSE as quality criteria—these parameters can be seen more as tuning variables rather than strictly hydrogeological parameters.

L680: As I stated before, I miss a part of the discussion about inversion convergence (RMSE and/or chi2) comparison between the inversion schemes in order to agree with this sentence

As noted in the comment above, a direct comparison of the error metrics between PJI-AR and PJI-GM for all profiles is not directly possible due to the fact that we did not systematically test (loop over) a large array of parameter values for the PJI-AR version of the model. A 1:1 comparison is given in Table 1 of this document for the four representative profiles chosen in the manuscript.

Figure A3: Fit the text to the shape within the figure

We will change the text to match the figure shapes.

Table A1: Increase font size; About RMSE: See comment for the table below + for consistency you should add also the chi2 measure

We increased the font size of this table and added the Chi2 values as an additional column.

Table A2: Increase font size; Of which inversion scheme? PJI-GM?; Maybe here you could add columns to compare the different methods misfits (individual, PJI-GM and PJI-AR)

Both Tables A1 and A2 show the filtering meta data for the individual data inversions, not the PJI results. Therefore, it is neither PJI-GM nor PJI-AR. We increased the font size of this table as well .

Additionally, we have now correctly referenced the Figure numbers, where those were marked as (?). We also corrected the references that were not correctly coded in the Latex file (also marked with (?)). Thanks a lot for pointing those out! We also removed parentheses for the years where this was suggested by the reviewer.

References:

Lebedeva, L. S., С., Л. Л., Kapitsa, V. Р., П., К. В., Takibaev, Z. D., Д., Т. Ж., Goncharenko, V. V., В., Г. В., Lytkin, V. M., М., Л. В., Kamalbekova, A. N., & H., K. A. (2024). On the influence of rock glacier dynamics on the runoff in basin of the Ulken Almaty (Bolshaya Almatinka) River, Northern Tien Shan. *Lëd i Sneg*, *64*(1), 54–65. https://doi.org/10.31857/S2076673424010041

Scherler, M., Hauck, C., Hoelzle, M., Stähli, M., & Völksch, I. (2010). Meltwater infiltration into the frozen active layer at an alpine permafrost site. *Permafrost and Periglacial Processes*, *21*(4), 325–334. https://doi.org/10.1002/PPP.694

Shokri, N., Hassani, A., & Sahimi, M. (2024). Multi-Scale Soil Salinization Dynamics From Global to Pore Scale: A Review. *Reviews of Geophysics*, *62*(4), e2023RG000804. https://doi.org/10.1029/2023RG000804

---

## Author Comment (AC2)

**Reviewer 1 comments and author answers:**

The paper describes a novel approach to permafrost PJI tested in different sites of Tien Shan and Pamir. The paper is well written, with rigorous description of the methodologies adopted. Conclusions are fully supported by the relevant results. The only criticism I have is about the paper length ( 52 pages) that makes the manuscript hard to read. In my opinion  the authors jointed 2 works that may be separated in 2 different contributions helping the reading: a work about the relevant permafrost characterisation of the remote studied areas, and a work about the novel PJI-GM approach and its comparison with the more common PIJ-AR. I obviously leave to the editor the decision about suggesting the splitting or not.

Dear Jacobo. Thank you for your positive feedback and thoughtful suggestion regarding the manuscript length. We understand the concern, but we believe splitting the paper would reduce its impact. The permafrost characterization alone might not attract enough interest, and keeping the work unified allows us to better connect the data with the methodological advancements. To improve readability, we will revise the manuscript to remove redundancy as suggested and make it more concise while keeping the content intact. We hope this addresses your concerns.

I suggest to sum up the discussion avoiding some repetition as in ln540, and to insert before the most relevant findings (e.g. Ln625-635) , and the important landforms / ice content relations.

We shortened the discussion, which is now about 30 lines shorter than in the original manuscript.

I noted some typo that need corrections:

LN 472 Sentence about support of  higher standard deviation not clear

Thank you for pointing out the unclear sentence on LN 472. We have reviewed the text and clarified the statement regarding the higher standard deviation:

This is also quantitatively supported by the higher standard deviations for the PJI-GM (see Fig. 12 1a vs. 1b), indicating greater variability in the tomograms. Higher standard deviations suggest the PJI-GM model captures more heterogeneous subsurface features.

LN 531, 533, 556 typo figures numbers

Thank you for noticing. We have corrected these errors.

---

## Author Comment (AC3)

**Reviewer 3 comments and author answers:**

This paper employs the Petrophysical Joint Inversion (PJI) method, combining the Geometric Mean Model and Clustering approach to quantify the ground ice content of mountain permafrost across different landforms. It compares these results with those obtained using Conventional Electrical Resistivity Tomography (ERT) Inversion and Archie's Law (PJI-AR) methods, evaluating the applicability of PJI-GM in mountain permafrost under various landforms. This study fills a data gap regarding the extent of ground ice in the Tien Shan and Pamir regions (Central Asia) and analyzes the respective advantages and disadvantages of PJI-GM and PJI-AR in mountain permafrost areas. The research is at the forefront of the field, the overall logic of the paper is clear, and I recommend acceptance after necessary revisions. Below are specific suggestions for modifications:

- Line 202: The authors mentioned using a modified PJI approach, which appears to refer to the PJI-GM method. However, in the introduction, the authors state that Mollaret et al. (2020) proposed the PJI-GM method (Line 107). It is unclear what are the modifications compared to Mollaret et al. (2020).

Thank you for pointing this out. We agree that the term modified might be misleading in this context, and we will remove it to avoid confusion. To clarify, Mollaret et al. (2020) primarily presented a proof of concept, demonstrating the use of the Geometric mean model within PJI for well-characterized profiles with known ground truth. In contrast, our study extends this approach by systematically testing its performance across a variety of conditions, including sites where no independent ground truth is available. Additionally, our implementation includes iterative inversion loops and cluster analysis, which allow for a more refined assessment of the method's applicability. By applying PJI-GM in these new settings, we aim to evaluate its robustness and potential for broader geophysical applications beyond the proof-of-concept stage.

- Line 292: Although authors mentioned defining the zone of interest (ZOI) for each profile according to the method of Hilbich et al. (2022) to calculate the potential ground ice content, I suggest that the authors provide a detailed explanation of how the ZOI is defined, as the extent of the ZOI directly affects the subsequent calculation of ground ice content for each profile.

Thank you for your suggestion. We acknowledge that the definition and delineation of the Zone of Interest (ZOI) involve a certain degree of subjectivity. However, we tested a range of ZOI extents for each profile and found that, in general, the variations had only a minor impact on the final ground ice content calculations. The ZOIs are, furthermore, shown in the Annexe Figure A2.

To improve clarity, we added a more detailed explanation of how the ZOIs were defined:

*To define the ZOIs, we typically selected a zone below the active layer that extends horizontally across as much of the profile as possible within the area where frozen conditions are expected. The depth and width of the ZOI was adjusted to each profile's resolution capacity in the relevant depth range to ensure a representative selection..*

- Line 294: The term "zone of interest" seems to refer to Figure A2 rather than Figure A1.

You are correct, thank you for noticing. We changed the manuscript accordingly.

- Line 313: "(Rücker et al., 2017)" should be changed to "Rücker et al. (2017)".

Thanks for noticing, we changed the citation accordingly.

- Line 363: "Figure A" should refer to "Figure A3," right?

Yes, thanks again for noticing, we changed the reference accordingly.

- The subfigure numbering format in all figures within the paper is inconsistent. Some use lowercase letters (a., b., c., d.), while others use uppercase letters (A, B, C, D).

We now use lowercase letters for all figures.

- Figure 5a: The y-axis is missing a label, and the legend of the colorbar could be adjusted slightly to the right.

We added a label for the y-axis.

- Figure 8: What do the blue dotted lines on the surface represent? I did not find an explanation in the legend.

The blue dotted lines on the surface indicate the location of the electrodes. We added this to the caption:

Examples of interpreted ERT profiles of different landforms. (a) rock glacier with high resistivities below an active layer of about 4 m; (b) moraine, high resistivities may point to buried glacier ice; (c) Rock glacier and fine grained sediments (d) talus slope; (e) fine-grained sediments; (f) fine-grained sediments, where a borehole confirmed saturated ground ice conditions in uppermost layers. The blue dotted lines on the surface indicate the location of the electrodes.

- Figure 11: A label indicating depth should be added to the y-axis.

We added the y-axis label.

- Figure 13: Similarly, a label indicating depth should be added to the y-axis.

We added the y-axis label.

- Lines 531, 533, and 556: The references to figures in the text are incorrect.

This was also indicated by the other reviewers and the correct figure references were added.

- Line 190: The extra question mark seems to indicate an incorrect citation?

Yes, we added the correct citation. Thanks for pointing this out.